# Gut Microbiota-Targeted Therapeutics for Metabolic Disorders: Mechanistic Insights into the Synergy of Probiotic-Fermented Herbal Bioactives

**DOI:** 10.3390/ijms26125486

**Published:** 2025-06-07

**Authors:** Yue Fan, Yinhui Liu, Chenyi Shao, Chunyu Jiang, Lijuan Wu, Jing Xiao, Li Tang

**Affiliations:** 1Department of Microecology, College of Basic Medical Science, Dalian Medical University, Dalian 116041, China; yfan_0220@163.com (Y.F.); yhliu_dl@163.com (Y.L.); shaocy@dmu.edu.cn (C.S.); 19861558290@163.com (C.J.); wulijuan939@126.com (L.W.); 2Department of Oral Pathology, College of Stomatology, Dalian Medical University, Dalian 116041, China

**Keywords:** gut microbiota, metabolic diseases, traditional Chinese medicine, probiotics, fermentation

## Abstract

Gut microbiota dysbiosis is intricately linked to metabolic disorders such as obesity, type 2 diabetes mellitus (T2DM), hyperlipidemia, and non-alcoholic fatty liver disease (NAFLD). Traditional Chinese medicine (TCM), particularly when combined with probiotic fermentation, offers a promising therapeutic strategy by modulating microbial balance and host metabolism. This narrative review synthesizes current research on probiotic-fermented herbal bioactives, focusing on their mechanisms in ameliorating metabolic diseases. Probiotic and bioactive compounds (e.g., berberine, polysaccharides) are highlighted for their roles in enhancing intestinal barrier function, regulating microbial metabolites like short-chain fatty acids (SCFAs), and reducing inflammation. Fermentation techniques improve the bioavailability of TCM components while reducing toxicity, as seen in fermented *Salvia miltiorrhiza* and *Rhizoma Coptidis*. Despite promising results, challenges include the complexity of microbiota–host interactions and variability in TCM standardization. Future directions emphasize integrating multi-omics technologies and personalized approaches to optimize probiotic-fermented TCM therapies. This review underscores the potential of combining traditional herbal wisdom with modern biotechnology to address metabolic disorders, which pose significant global health challenges, through a “gut microbiota–metabolism” axis. Emerging evidence highlights the critical role of gut microbiota dysbiosis in the pathogenesis of these conditions. TCM has shown promise in modulating gut microbiota to restore metabolic homeostasis. This review synthesizes current research on TCM-derived interventions, such as herbal compounds, probiotics, and fermentation techniques, that target gut microbiota to ameliorate metabolic disorders. We discuss mechanisms of action, including prebiotic effects, enhancement of intestinal barrier function, and regulation of microbial metabolites, while addressing the limitations and future directions of TCM-based therapies.

## 1. Introduction

Metabolic disorders, including obesity, T2DM, hyperlipidemia, and NAFLD, represent a global health crisis, affecting over 1.9 billion adults and contributing to significant morbidity and mortality worldwide [1]. These conditions are intricately linked to dysregulated energy metabolism, chronic inflammation, and insulin resistance, with emerging evidence highlighting the gut microbiota as a pivotal mediator of metabolic homeostasis [2]. The human gut microbiota, a complex ecosystem of trillions of microorganisms, encodes over 3 million genes—far exceeding the human genome—and exerts profound influence on nutrient metabolism, immune regulation, and endocrine signaling [3]. Dysbiosis of this microbial community, characterized by reduced diversity, overgrowth of pro-inflammatory pathogens, and depletion of beneficial taxa (e.g., *Akkermansia muciniphila* and *Bifidobacterium*), is now recognized as a hallmark of metabolic diseases [4,5].

Conventional therapies for metabolic disorders, such as lifestyle interventions, Glucagon-like peptide-1 (GLP-1) agonists, and sodium–glucose transport protein 2 (SGLT2) inhibitors, offer partial relief but are limited by side effects, high costs, and variable efficacy [6,7]. This has spurred interest in gut microbiota-targeted strategies, including probiotics, prebiotics, and fecal microbiota transplantation (FMT), to restore microbial balance and improve metabolic outcomes [8]. Among these approaches, TCM has gained attention for its multi-target, holistic mechanisms. TCM herbs, enriched with polysaccharides, flavonoids, and alkaloids, modulate gut microbiota composition, enhance intestinal barrier integrity, and regulate microbial metabolites like SCFAs [9]. However, the bioavailability and toxicity of certain TCM components, such as aconitine in *Aconitum* species, hinder their clinical utility [10].

Probiotic fermentation has emerged as a transformative technology to overcome these limitations. By leveraging strains like *Lactobacillus* and *Bifidobacterium*, fermentation enhances the bioactivity of herbal compounds, degrades toxic alkaloids, and generates novel metabolites with anti-inflammatory and antioxidant properties [11]. For example, fermented *Salvia miltiorrhiza* exhibits increased tanshinone bioavailability, which reduces hepatic lipid accumulation in NAFLD, while fermented *Rhizoma Coptidis* converts berberine into more absorbable forms, improving insulin sensitivity [12,13]. This synergy between probiotics and herbal bioactives aligns with the “gut microbiota–metabolism” axis, offering a natural, multidimensional therapeutic strategy [14].

This narrative review synthesizes current research on probiotic-fermented TCM interventions for metabolic disorders, with three primary objectives: (1) to elucidate the mechanisms by which fermentation enhances the efficacy and safety of herbal bioactives; (2) to evaluate the role of probiotic-fermented TCM in modulating gut microbiota composition, SCFA production, and immune responses; and (3) to address challenges in standardization, scalability, and personalized application. We conducted a systematic literature search using Web of Science, Ovid Medline, PubMed and Google Scholar (2019–2025), focusing on peer-reviewed studies investigating probiotic strains, fermentation techniques, and metabolic outcomes. Key search terms included “traditional Chinese medicine”, “gut microbiota”, “metabolic diseases”, “type 2 diabetes”, “non-alcoholic fatty liver disease”, “gout”, “hyperuricemia” and “obesity” (Figure 1).

The rationale for this review is underscored by the urgent need to integrate traditional medical wisdom with modern biotechnology. While preclinical studies demonstrate promising results, clinical translation remains hampered by heterogeneity in TCM formulations and a lack of mechanistic clarity [15].

Furthermore, the dynamic interplay between microbial metabolites (e.g., SCFAs, bile acids) and host pathways requires advanced multi-omics technologies to unravel species-specific contributions [16]. By addressing these gaps, this work aims to advance probiotic-fermented TCM as a precision medicine tool, tailored to individual microbial and metabolic profiles.

## 2. Metabolic Diseases

### 2.1. Epidemiology and Clinical Significance of Metabolic Diseases

Metabolic diseases, a group of diseases characterized by metabolic abnormalities, include hyperlipidemia, T2DM, obesity, NAFLD, gout, and osteoporosis [17]. Such diseases have emerged as a global health crisis. According to data from the World Health Organization and the International Diabetes Federation’s IDF Diabetes Atlas 11th Edition 2025, the current affected population comprises 2.2 billion obese adults and 589 million individuals with diabetes, posing a significant challenge to public health systems.

Recent studies have demonstrated that the gut microbiota and its bioactive metabolites play a pivotal role in systemic metabolic regulation via the “gut-metabolism axis” [18]. Consequently, targeted modulation of the gut microbiota and its metabolic network has become a promising therapeutic strategy for metabolic diseases.

#### 2.1.1. Obesity

Obesity has become a major global public health challenge, and its prevalence has increased dramatically in recent decades. The World Obesity Atlas 2024 report predicts that by 2035, over 750 million children will be classified as overweight or obese, representing two out of every five children globally. Data from the Global Burden of Disease Study (GBD) indicate that there were 5 million obesity-related deaths in 2019, making obesity the leading cause of mortality among metabolic diseases.

Recent research indicates that the gut microbiota significantly regulates the pathogenic mechanisms of obesity. Specifically, gut microbiota dysbiosis, such as an abnormally elevated *Firmicutes*-to-*Bacteroidetes* ratio, is significantly associated with enhanced dietary energy absorption efficiency and adipose tissue accumulation in the host [19]. Notably, the reduced abundance of symbiotic bacterial communities capable of producing SCFAs, such as *Faecalibacterium prausnitzii* and *Akkermansia muciniphila*, disrupts the regulation of energy homeostasis signals, thereby impairing the organism’s energy balance mechanism [20]. Based on these findings, the targeted modulation of the bidirectional communication pathway between the gut and brain has emerged as a novel intervention technique for the prevention and treatment of obesity and associated metabolic disorders.

#### 2.1.2. T2DM

The prevalence of T2DM has increased by more than 1.5% per year over the past 20 years, and according to the results of the GBD study published in *The Lancet*, there will be 529 million people with diabetes globally in 2021. By 2050, there will likely be 1.31 billion diabetics worldwide [21]. The occurrence and development of T2DM are mainly the result of the combined effects of genetics, environment, lifestyle and epigenetics [22,23].

The traditional theory holds that insulin resistance and islet β-cell dysfunction are its core mechanisms. In recent years, emerging theories have revealed their intricate mechanics. Theories such as epigenetic regulation [24], chronic low-grade inflammation [25], and mitochondrial dysfunction [26] have provided new perspectives for the treatment of T2DM. The gut microbiota–metabolic axis has garnered significant attention. Dysregulation of gut microbiota exacerbates insulin resistance and glycolipid metabolic diseases by mechanisms including diminished production of SCFAs, compromised intestinal barrier integrity, and disruption of bile acid metabolism [27].

#### 2.1.3. Hyperlipidemia

Hyperlipidemia is characterized by elevated levels of low-density lipoprotein cholesterol (LDL-C), reduced high-density lipoprotein cholesterol (HDL-C), and increased triglyceride (TG). It represents a major risk factor for various cardiovascular diseases [28]. Over the past three decades, the prevalence of hyperlipidemia has increased significantly worldwide. The ranking of hyperlipidemia among global risk factors for mortality increased from 15th in 1990 to 11th in 2007 to 8th in 2019 [29]. Elevated LDL-C and reduced HDL-C levels are particularly associated with an increased risk of atherosclerosis and cardiovascular events. Studies have shown that modifications in the gut microbiota play a crucial role in developing hyperlipidemia and its related metabolic disorders. The dysbiosis of the gut microbiota, characterized by changes in microbial composition and function, correlates with changes in lipid metabolism and an elevated risk of cardiovascular illnesses [30].

NAFLD is becoming increasingly prevalent, with an estimated annual incidence rate of 0.83% [1]. By 2030, the global number of NAFLD patients is projected to reach approximately 101 million [16]. NAFLD is commonly associated with hyperlipidemia, sharing common pathophysiological mechanisms such as insulin resistance, lipid metabolism disorders, and alterations in the gut microbiota. Metabolites originating from the gut microbiota, such as secondary bile acids, may affect lipid control. Alterations in gut microbiota composition can modify the bile acid profile, consequently influencing lipid absorption, cholesterol balance, and systemic inflammation [31]. The high incidence and mortality rates associated with hyperlipidemia and NAFLD emphasize the urgent need to study suitable therapies and interventions.

#### 2.1.4. Hyperuricemia and Gout

Gout is primarily caused by chronically elevated serum uric acid levels, a condition known as hyperuricemia—sodium urate crystals deposit in joint tissues, triggering recurrent and pronounced acute inflammatory responses. Gout is a metabolic disorder distinguished by painful arthritis, particularly affecting the joints of the lower limbs.

The GBD Study indicates that the global population of gout patients increased from roughly 22 million in 1990 to about 53.9 million in 2019, reflecting a 63.44% rise in incidence rate. The incidence rate is becoming prevalent among young individuals.

The gut microbiota plays a pivotal role in uric acid metabolism and excretion. Certain bacteria within the gut microbiota can metabolize purines and uric acid, thereby influencing serum uric acid concentrations [32]. Zhao et al. [33] found that the gut microbiota composition in gout patients is significantly distinct from that in healthy individuals. Alterations in bacterial genera, such as *Faecalibacterium prausnitzii* and *Bacteroides uniformis*, may influence uric acid homeostasis. Additionally, Lv et al. [34] revealed that an increased abundance of inflammation-related microbiota and elevated uric acid levels in hyperuricemia are linked to intestinal barrier dysfunction, which disrupts the host–microbiota crosstalk.

### 2.2. Treatment of Metabolic Diseases

#### 2.2.1. Lifestyle Interventions

Current interventions for metabolic diseases primarily involve dietary modifications [35], appropriate exercise [36], and adjustments in work and rest patterns. Regular exercise and physical activity are essential for improving metabolic diseases, contributing to increased lifespans, preventing aging and associated diseases, and serving as a critical intervention for preventing common metabolic disorders and enhancing both physical and mental health [37,38]. Studies have shown that exercise can control blood glucose in patients with T2DM [39]. Exercise inhibits the key enzyme in liver gluconeogenesis by activating AMP-activated protein kinase (AMPK), thereby enhancing liver insulin sensitivity. Moreover, exercise activates AMPK via adrenergic signaling, which promotes adipose tissue lipolysis, reduces fat accumulation, and alleviates NAFLD [40].

Exercise has been demonstrated to enhance the variety of the gut microbiota. Studies have shown that the gut microbiota diversity in athletes is markedly greater than that in sedentary individuals, particularly enriched with beneficial bacterial taxa such as *Akkermansia muciniphila* and *Prevotella* spp. Endurance exercise can diminish the pro-inflammatory bacteria *Proteobacteria*, consequently enhancing metabolism and reducing the risk of metabolic disorders [41,42].

Similarly, inappropriate lifestyle, disruption of circadian rhythms [43], and food consumption at inappropriate times can disrupt the temporal coordination of metabolism and physiology [44], thereby promoting metabolic diseases. Adjusting the work routine has become one of the therapeutic approaches for metabolic diseases.

#### 2.2.2. Drug Treatment

Several effective medications have been developed to lower triglycerides, blood glucose, and other metabolic biomarkers in recent years. The pharmaceutical treatment of metabolic diseases has become a crucial technique [45], including agents such as insulin sensitizers, lipase inhibitors, xanthine oxidase inhibitors, and farnesoid X receptor (FXR) agonists. However, they still have some limitations due to the possibility of gastrointestinal discomfort, hypoglycemia, hepatic dysfunction, and weight gain, as well as the occurrence of adverse effects such as rebound, inflammation, and impaired excretion [28,46]. Meanwhile, some novel therapeutic tools include SGLT2 inhibitors, GLP-1 receptor agonists and other novel glucose-lowering drugs. GLP-1 is an enteric insulin-secreting molecule used in the treatment of T2DM because of its properties, such as causing central appetite suppression, protection of pancreatic islet β-cells, and cardiovascular protective effects [47].

GLP-1R is a G-protein-coupled receptor that is widely found on the surface of various human cells and mainly mediates the physiological and pharmacological effects of GLP-1, which regulates blood glucose levels and lipid metabolism in humans. This receptor and its agonists show important therapeutic potential for the treatment of metabolic diseases [48,49,50].

Clinical studies demonstrate that GLP-1 receptor agonists improve blood glucose regulation and favorably affect the gut microbiota. They may promote the proliferation of beneficial bacteria, such as *Akkermansia muciniphila*, which is associated with improved metabolic outcomes [51].

## 3. The Correlation Between Gut Microbiota and Metabolic Diseases

### 3.1. The Fundamental Role of Gut Microbiota in Disease and Health

The human genome has approximately 23,000 genes, but the human commensal microbiome encodes over 3 million genes, with metabolites influencing several host activities, hence impacting the host’s metabolic balance, phenotypic traits, and overall health [3]. As the largest microbial community within the human body, the gut microbiota serves a critical role in maintaining a harmonious host-microbe relationship. It’s an detailed ecosystem of trillions of bacteria, viruses, archaea, and fungi, all deeply interconnected with the host’s metabolic health and disease risk [2,52]. Interestingly, the genetic diversity within the human microbiota exceeds that of our own genome by roughly 130 times. These microorganisms are essential for key body processes like nutrient breakdown, immune regulation, and neuroendocrine function, employing complex regulatory mechanisms that support the host’s well-being [53].

The gut microbiota is affected by the method of birth, infant nutrition, lifestyle, medications, and host genetics. It is crucial for maintaining immunological homeostasis and processes [54], food digestion, interactions with the enteroendocrine system and neural signaling [55], as well as influencing therapeutic efficacy and toxicity [56].

#### 3.1.1. Therapeutic Approaches for Diseases of the Digestive System

Inflammatory bowel disease (IBD) and irritable bowel syndrome (IBS) are common gastrointestinal disorders whose pathogenesis is closely linked to intestinal dysbiosis. For instance, Crohn’s disease (a subtype of IBD) is associated with reduced microbial diversity and significant ecological imbalance, which correlate with intestinal mucosal damage [57,58]. In IBS patients, abnormal gut microbial metabolism manifests as increased gas-producing bacteria (e.g., methanogens) leading to abdominal bloating, while reduced SCFAs impair intestinal motility. Furthermore, dysbiosis exacerbates abdominal pain and bowel irregularities by disrupting neurotransmitters such as 5-hydroxy tryptamine through the gut–brain axis. *Bifidobacterium* and *Lactobacillus* are the most widely reported probiotic strains effective in treating IBD and IBS [59,60]. Butyrate neutrophil functions and ameliorates mucosal inflammation in IBD [61]. Gamma-aminobutyric acid (GABA), synthesized by gut microbes such as *Lactobacillus*, reduces intestinal smooth muscle tone via the gut–brain axis, thereby alleviating abdominal pain and diarrhea in IBS [62].

In addition, gut microbes and their metabolites may affect the efficacy of various cancer therapies, including chemotherapy, radiation, and immune checkpoint inhibitors, while also mitigating adverse effects such as mucositis, colitis, and neurotoxicity associated with cancer treatment [63]. The gut barrier–microbiota axis maintains intestinal homeostasis through tripartite regulation of metabolic, immune, and physical barriers. Its dysregulation serves as a central mechanism underlying diseases such as IBD and IBS. Consequently, modulating this axis via probiotics, dietary interventions, and microbiota transplantation has emerged as a pivotal focus in clinical research.

#### 3.1.2. Therapeutic Approaches for Diseases of the Neurological System

Studies have shown a close association between gut microbiota and a variety of neuropsychiatric disorders. The gut microbiota may affect brain function and behavior via the gut–brain axis, and its metabolites potentially play a role in neurotransmitter control, providing new research directions for the pathogenesis and treatment of these diseases [64].

The development of CNS disorders can be facilitated when the gut microbiota undergoes structural disturbances, abnormal microbial metabolism, gut barrier disruption and inflammation. Regulating the balance of gut microbiota, such as intermittent fasting, prebiotics, probiotics, and FMT may be a good option for recovering from CNS disorders.

*Lactobacillus rhamnosus* has been demonstrated in multiple studies to exert beneficial effects on anxiety and stress models. The underlying mechanisms involve modulating GABA expression in the central nervous system, reducing stress hormone corticosterone levels, regulating monoamine neurotransmitters (e.g., norepinephrine, dopamine, and serotonin) in the prefrontal cortex, and influencing activity in emotion-related brain regions such as the amygdala and prefrontal cortex via vagal afferent fibers that target the nucleus tractus solitarius (NTS). Notably, although significant efficacy has been demonstrated in animal models, human trials exhibit substantial inter-individual variability in therapeutic outcomes.

Regulating the balance of gut microbiota may promote recovery from CNS disorders by restructuring the intestinal microecological balance and modulating the biosynthesis and functional activity of key microbial metabolites, such as SCFAs, lipopolysaccharide (LPS), and GABA through multi-targeted regulatory mechanisms [65]. Given the significant individual variations observed in human trials, although these strategies demonstrate exciting therapeutic potential, their practical clinical application still requires further exploration.

#### 3.1.3. Therapeutic Approaches for Diseases of the Urinary System

##### Urinary Tract Infections

Some common urinary tract diseases, such as urinary tract infections, are common complications in kidney transplant recipients [66]. Chio et al. [67] demonstrated that the gut microbiota is associated with urinary tract colonization of uropathogenic intestinal reservoirs during recurrent urinary tract infections, that uropathogenic intestinal reservoirs can asymptomatically colonize the intestinal and urinary tracts, and that gut microbiota proliferates in patients with urinary tract colonization, suggesting that transhabitat migration of uropathogenic intestinal reservoirs is an important mechanism for recurrent urinary tract infections.

Probiotics do not have a direct therapeutic effect on urinary tract infections (UTIs). However, studies suggest that certain strains can competitively adhere to the uroepithelium, blocking pathogen invasion and reducing the risk of UTI recurrence. Additionally, probiotics may enhance local immune defenses in the urinary system by activating immune cells to secrete antimicrobial peptides and other protective substances.

##### Chronic Kidney Disease and Urological Cancers

Studies have shown that there is a strong association between gut microbiota and urological cancers, as known using Mendelian randomization analysis [68]. Several other studies have shown that chronic kidney disease (CKD) is associated with changes in the gut microbiota composition. For example, the gut microbiota produces metabolites, SCFAs, which may have an impact on kidney health [69]. Elevated urea levels during CKD expedite renal damage, resulting in modifications to the gut microbiota. This, in turn, enhances the synthesis of gut-derived toxins and disrupts the intestinal epithelial barrier, thereby impacting the advancement of CKD [70]. Toxin-producing gut microbiota can also exacerbate the clinical prognosis of end-stage renal disease through the production of deleterious metabolites, suggesting that uremic toxicity can be reduced in patients by modulating the gut microbiota to improve survival or reduce the need for dialysis therapy and renal transplantation resources [71].

Research has shown that specific probiotics may slow the progression of CKD. A Phase 1 study involving 62 patients with stage 3–5 CKD demonstrated that supplementation with *Lactobacillus casei* Zhang significantly slowed the decline in renal function. Additionally, studies suggest that certain probiotics can enhance immunotherapy efficacy. For example, a Phase 1 clinical trial revealed that combining the probiotic *Clostridium butyricum* CBM588 with immune checkpoint inhibitors (ICIs) significantly improved the objective response rate (ORR) in patients with metastatic renal cell carcinoma.

All of the above research elements demonstrate the development of new therapeutic tools and strategies by exploring the effects of gut microbiota on the host. Probiotics demonstrate strong therapeutic potential in alleviating kidney injury, preventing UTIs, or reducing the risk of UTI recurrence. Since the influence of gut microbiota on the body’s metabolism is multidimensional, it is possible to provide ideas for the study and treatment of diseases by exploring different modes of action.

### 3.2. Interactions Between Gut Microbiota and Metabolic Diseases

#### 3.2.1. Alterations in Gut Microbiota at the Onset of Metabolic Diseases

The composition and functional activity of gut microbiota significantly regulate the start, progression, and remission of metabolic disorders. Gut microorganisms not only affect the host’s nutritional metabolism but are also intricately linked to the host’s physiological processes via their metabolites.

Multiple studies have identified the gut microbiota as a central player in the pathophysiology of metabolic syndrome, with its dysbiosis closely linked to obesity, insulin resistance, and disrupted lipid homeostasis. Restoring microbial balance to improve health outcomes has become a key target for numerous therapeutic interventions [72].

##### Alterations in Gut Microbiota in Obesity

It has been found that gut microbiota diversity is usually significantly reduced in individuals with metabolic diseases. For obese individuals, the ratio of the Firmicutes to the Bacteroidetes in their gut microbiota was modified, exhibiting an increase in the former and a decrease in the latter [73,74]. Some bacteria in the phylum *Firmicutes* possess an enhanced capacity to extract energy from food, facilitating their hosts’ absorption of additional calories from their diet, potentially resulting in energy buildup and obesity [75]. In addition, bacteria within the *Firmicutes* phylum ferment dietary fiber and convert it into SCFAs, and these metabolites may affect energy metabolism and fat storage in the host, which may contribute to the development of obesity [5]. Some endotoxin-producing Gram-negative bacteria (e.g., *Enterobacteriaceae*) may be increased in the gut of obese individuals. The cell wall component of these bacteria, LPS, can enter the circulation and trigger chronic low-grade inflammation [76].

Obesity, particularly abdominal obesity (central obesity), is a central component of metabolic syndrome (MetS). The accumulation of visceral adipose tissue (VAT) directly impairs insulin signaling through dysregulated secretion of adipokines such as leptin and adiponectin, leading to insulin resistance. This cascade further drives glucose metabolism dysregulation, dyslipidemia, and hypertension, collectively defining the core features of MetS [77]. Given the central role of obesity, timely interventions targeting obesity are critical to alleviating the associated disease burden.

##### Alterations in Gut Microbiota in T2DM

In patients with T2DM, changes in gut microbiota are closely related to the development of insulin resistance. Decreased populations of beneficial bacteria, such as *Bifidobacterium* spp. and increased populations of some conditionally pathogenic bacteria, such as certain *Clostridium* spp. and *Enterobacteriaceae*, can lead to impaired intestinal barrier function [78]. In addition, reductions in specific bacterial species, such as *Akkermansia muciniphila*, are associated with reduced insulin sensitivity [4]. *Streptococcus* spp. and *Lactobacillus* spp. that produce lactic acid are found in higher proportions, whereas *Blautia* spp. and *Anaplasma* spp. that produce SCFAs are lower in diabetic than non-diabetic individuals [79]. All of the above studies have demonstrated that changes in gut microbiota affect glycemic control in patients with T2DM.

Gut microbiota dysbiosis reduces SCFA levels, leading to diminished activation of GPR41 and GPR43, which promotes intestinal inflammation and insulin resistance. Concurrently, reduced bile acid levels result in weakened activation of FXR and TGR5, suppressing insulin secretion and impairing insulin sensitivity. Compromised intestinal barrier function further activates Toll-like receptor (TLR) and mitogen-activated protein kinase (MAPK) signaling pathways, inducing endotoxemia and exacerbating inflammatory responses and insulin resistance [80].

##### Alterations in Gut Microbiota in Hyperlipidemia

Recent studies have shown that the imbalance of gut microbiota homeostasis is an important environmental factor driving the occurrence of hyperlipidemia. Patients with hyperlipidemia usually consume foods that are high in fat and cholesterol. This dietary structure can alter the nutritional environment in the intestines and affect the gut microbiota. This disorder of flora can cause damage to the intestinal barrier [30]. In addition, alterations in the metabolite profile of the microbiota can interfere with the signal transduction of FXR and liver X receptor (LXR), forming a vicious cycle of lipid metabolism [81].

Huang et al. [82] found that in a hyperlipidemia model, pro-inflammatory bacteria such as *Bilophila*, *Turicibacter*, and *Colidextribacter* were significantly increased, showing a positive correlation with serum levels of total cholesterol (TC), LDL-C, and inflammatory cytokines. The pathological mechanisms of hyperlipidemia are closely linked to gut-liver axis dysregulation.

FXR, the primary nuclear receptor for bile acids, exhibits reduced activity due to gut microbiota dysbiosis-induced depletion of secondary bile acids (e.g., deoxycholic acid, DCA). This leads to derepression of sterol regulatory element-binding protein 1c (SREBP-1c), resulting in aberrant activation of hepatic lipid synthesis pathways. Concurrently, gut microbiota-derived metabolites (e.g., ethanol, LPS, SCFAs) activate pro-inflammatory signaling (e.g., IL-6/STAT3) and oxidative stress, directly upregulating lipogenesis-related genes [83].

##### Alterations in Gut Microbiota in NAFLD

In patients with NAFLD, the gut microbiota shows significant dysbiosis, with changes characterized by a decrease in beneficial bacteria (e.g., *Clostridium perfringens*) and an increase in pro-inflammatory pathogenic bacteria (e.g., *Klebsiella pneumoniae*) [84].

Alterations in microbiota result in compromised intestinal barrier function, facilitating the entry of bacterial metabolites (e.g., endotoxins) into the liver, which incites hepatic inflammation and lipid buildup. Additionally, metabolites produced by certain bacterial strains can promote disease progression. For example, *High-Alcohol-Producing Klebsiella Pneumoniae* causes an elevation in ethanol production [85]. For example, there may be an elevation in ethanol production by the gut microbiota, which can transit to the liver via the portal vein and facilitate hepatic steatosis. At the same time, dysbiosis also interferes with bile acid metabolism, leading to increased liver burden and promoting the progression of NAFLD to more serious liver diseases, such as non-alcoholic steatohepatitis (NASH) and cirrhosis [86].

Fecal bile acid testing in patients with NAFLD revealed that the bacteria *Escherichia* spp. were involved in taurine and glycine metabolism, while *Cholera* spp. and *Erythrobacter* spp. were significantly enriched in the feces of patients with NASH. *Cholera* spp. promote secondary bile acid production, exacerbating hepatic lipid deposition and inflammatory responses through activation of the FXR and Takeda G protein-coupled receptor 5 (TGR5) signaling pathways [87,88]. Patients with NAFLD exhibit a significantly higher ratio of *Firmicutes* to *Bacteroidetes* in their gut microbiota compared to those with NASH [89]. In NASH patients, key variables such as the serum bile acid profile and the hepatic gene expression pattern associated with increased bile acid synthesis are closely linked to alterations in the gut microbiota. These patients exhibit increased intestinal permeability and a decline in intestinal bacterial diversity, which may contribute to further liver injury [89,90].

Restoring microbial–host metabolic homeostasis through targeted modulation of the gut microbiota (e.g., probiotic interventions, fecal bacterial transplantation) has become an important strategy in the management of NAFLD.

##### Alterations in Gut Microbiota in Hyperuricemia and Gout

As microbiome research advances, there is growing evidence of a complex interaction between gut microbiota and gout. The structure of the gut microbiota in gout patients differs significantly from that of the healthy population. Research indicates that the diversity of gut microbiota is diminished in people with gout, while the prevalence of some specific microbial communities is either elevated or reduced [33]. Shao et al. [91] identified a heightened prevalence of conditionally pathogenic bacteria, such as increased levels of *Bacteroidetes*, *Corynebacteriaceae*, etc., and lower proportions of *Vibrio*, and *Ruminalococcaceae*, etc., by sequencing and comparing the fecal samples from the healthy group and the group of gout patients.

Chu et al. [92] found that the relative abundance of *Prevotella*, *Fusobacterium* and *Bacteroides* in gout increased. In contrast, the relative abundance of *Enterobacteriaceae* increased, and the number of butyrate-producing species decreased through metagenomic sequencing of stool samples from gout patients and healthy controls. LPS from Gram-negative bacteria enters the systemic circulation through a compromised intestinal barrier, activating the TLR4 signaling pathway in the liver and kidneys. This triggers oxidative stress and the release of inflammatory cytokines, ultimately suppressing the function of urate excretion-associated transporters while enhancing the activity of enzymes involved in uric acid production [93].

It was learned that the relative abundance of *Anaplasma*, *Prevotella* and *Clostridium* spp. increased in gout, whereas the relative abundance of *Enterobacteriaceae* and butyrate-producing species decreased [92], and the abundance of beneficial bacteria such as *Faecalibacterium prausnitzii* and *Bifidobacterium* spp. decreased [94]. In addition, gut microbiota participates in purine metabolism, influencing uric acid synthesis and excretion. *Escherichia coli* and *Aspergillus* spp. have been demonstrated to secrete xanthine dehydrogenase, an essential enzyme for the conversion of xanthine to uric acid [95], and the activity of xanthine oxidase may increase in the presence of an imbalance in the gut microbiota, leading to increased uric acid production.

#### 3.2.2. Regulation of Gut Microbiota for the Treatment of Metabolic Diseases

Modulating the gut microbiota has been identified as a viable approach for addressing metabolic illnesses. Dietary interventions, probiotic and prebiotic use [8], FMT [96], and pharmacological treatments can improve the composition of the gut microbiota, which in turn improves metabolic health and treats metabolic diseases.

##### Dietary Treatment

The diet is a crucial determinant in modulating the composition and functionality of the gut microbiota. It significantly influences the gut microbiota through a complicated network of interactions among nutrients, microbes, and hosts [97]. For example, the Mediterranean diet enhances the abundance of *Faecalibacterium prausnitzii* and *Roseburia* spp., shaping a gut microbiota enriched with functional capabilities in SCFA fermentation and dietary fiber degradation [98]. Increased dietary fiber consumption can promote the growth of beneficial bacteria, such as *Lactobacillus*, and increase the production of SCFAs, thereby improving insulin sensitivity [99]. Ketogenic diets are associated with a reduction in *Firmicutes* and Actinobacteria [100].

Gut microbiota can also produce SCFAs by utilizing polysaccharides. These essential microbial metabolites not only preserve intestinal barrier integrity but also modulate immune homeostasis via G protein-coupled receptor-mediated signaling pathways, playing a significant role in the prevention and treatment of metabolic diseases [101]. Fermentable fibers are recognized for their ability to selectively enrich beneficial gut microbiota and suppress pathogens, thereby improving metabolic imbalances [102]. However, a study revealed that low-fermentable fiber (LF) combined with FMT demonstrated superior efficacy compared to high-fermentable fiber (HF) in patients with severe obesity and metabolic syndrome, suggesting the need for deeper investigation into the application of fermentable fibers [103].

##### Probiotics

Probiotics, such as *Bifidobacteria* and *Lactobacilli*, contribute positively to the microecological balance of the host’s digestive tract, hence benefiting the host’s health and overall well-being. It mainly affects host health by regulating the gut microbiota and participating in the immune regulation of many diseases, thus improving gastrointestinal physiology.

Probiotic supplementation can also improve purine metabolism and lower blood uric acid levels. Some specific probiotics, such as *Bifidobacteria* and *Lactobacilli*, have attracted attention for their function in the treatment and alleviation of hyperuricemia [104,105]. In studies on hyperuricemic mice, *Limosilactobacillus reuteri HCS02-001* was found to ameliorate hyperuricemia by enhancing gastrointestinal barrier function to inhibit uric acid biosynthesis and promoting uric acid clearance through the upregulation of urate transporters [106].

Numerous clinical studies have explored the feasibility of probiotics in the treatment of T2DM. A meta-analysis by Tao et al. [107] revealed that probiotics significantly reduced fasting blood glucose, HbA1c and HOMA-IR in patients with T2DM. The specific strains reported to confer these benefits included *Lactobacillus acidophilus*, *Lactobacillus casei*, *Lactobacillus rhamnosus*, *Lactobacillus bulgaricus*, *Bifidobacterium bifidum, Bifidobacterium longum*, etc.

Li et al. [108] discovered that the *Lactobacillus gasseri RW2014* ameliorates hyperlipidemia by modulating bile acid metabolism and gut microbiota composition in rats. Whether there are differences in therapeutic efficacy between single-strain probiotics and multi-strain probiotics is under investigation.

In a study by Duseja A et al. [109], a high-concentration multi-strain probiotic formulation (including *Lactobacillus paracasei DSM 24733*, *Lactobacillus plantarum DSM 24730*, *Lactobacillus acidophilus DSM 24735*, *Lactobacillus delbrueckii* subsp. *Bulgaricus DSM 24734*, *Bifidobacterium longum DSM 24736*, *Bifidobacterium infantis DSM 24737*, *Bifidobacterium breve DSM 24732*, and *Streptococcus thermophilus DSM 24731*) was administered to patients with NAFLD. The intervention demonstrated significant improvements in liver histology, alanine aminotransferase (ALT) levels, and cytokine profiles compared to baseline or placebo groups.

In the clinical management of gout, probiotics can serve as an adjunct therapy. A probiotic formulation containing five strains (*Lactobacillus paracasei Zhang*, *Lactobacillus plantarum P-8*, *Lactobacillus rhamnosus Probio-M9*, *Bifidobacterium lactis Probio-M8*, and *Bifidobacterium lactis V9*) has been clinically proven to significantly enhance the efficacy of febuxostat in gout patients. The combination therapy resulted in greater reductions in serum uric acid levels, lower incidence of acute gout flares, and decreased serum creatinine levels, indicating improved renal function [110].

Overall, due to the small sample sizes of existing probiotic-based clinical trials, further large-scale, multicenter clinical trials are required to validate their therapeutic efficacy and evaluate potential adverse reactions.

##### Probiotics and Postbiotics Treatment

Prebiotics are soluble dietary fibers that gut bacteria can decompose into SCFAs, which serve as a primary energy source for intestinal epithelial cells and contribute to the preservation of intestinal barrier integrity [111]. Research indicates that prebiotics can selectively stimulate beneficial bacteria (e.g., *Bifidobacterium bifidum* and *Lactobacillus*) while improving intestinal epithelial barrier integrity through the modulation of tight junction protein expression [112]. It can exert a dual regulatory influence on the form and function of the gut microbial community.

Postbiotics are metabolites produced by beneficial bacteria following the metabolism of prebiotics or probiotic components, or cell wall-derived fractions released by bacterial cell lysis [112], which, according to the definition of the International Society for the Scientific Association of Probiotics and Prebiotics (ISAPP), refers specifically to “soluble functional factors secreted by living bacteria or released by bacterial lysis” [113]. These include vitamin B12, vitamin K, folic acid, LPS, enzymes, and SCFAs [114].

The production of postbiotics by gut microbiota is closely related to the host intestinal microenvironment, with research indicating that postbiotics may exert diverse effects, including enhancing intestinal barrier function and modulating systemic immune responses through interactions with intestinal epithelial cells. Furthermore, postbiotics possess anti-inflammatory, antibacterial, antioxidant, antihypertensive, and hypocholesterolemic effects, among others, and may be utilized in the management of associated metabolic disorders and diseases [115]. Lee et al. [116] found that *Lactobacillus plantarum* L-14 extracts (EPS) serve as an antioxidant in a mouse model of obesity induced by a high-fat diet, operating through the TLR2 and AMPK signaling pathways to inhibit the differentiation of adipose precursor cells to mature adipocytes, significantly alleviating obesity and related metabolic disorders. This finding provides a significant experimental foundation for utilizing postnatal metabolites in the management of metabolic disorders.

Postbiotic products have been shown to have a relatively weak ability to regulate gut metabolism or influence gene expression related to nutrient metabolism. Additionally, inactivation processes can impair the viability and immunomodulatory properties of probiotics. Furthermore, appropriate methods must be used to characterize and quantify the components and quantities of postbiotic products. These findings collectively suggest that the application of postbiotics necessitates further in-depth research to mitigate their potential risks [117].

##### FMT

FMT is an innovative therapeutic modality involving the transfer of a healthy microbiome sample to a host exhibiting a specific ecological imbalance. The goal is to recolonize the host gut with beneficial microorganisms [118] to restore the balance of the gut microbiota [72]. It has been included in the clinical practice guidelines for the treatment of recurrent *Clostridium difficile* infections. Furthermore, this therapy shows potential in treating certain metabolic diseases [119].

Wang et al. [120] transplanted fecal microbiota from healthy mice into diabetic mice and demonstrated that FMT restored the balance of the gut microbiota, increasing the abundance of beneficial bacteria such as *Firmicutes* and *Faecalibacterium prausnitzii*, while reducing the proportion of opportunistic pathogenic bacteria (e.g., *Bacteroides*). This microbial modulation consequently improved metabolic and immune functions, suggesting potential therapeutic implications for individuals with T2DM. Su et al. [121] demonstrated through a 90-day controlled open-label trial that FMT significantly reduced blood glucose levels in patients with T2DM and promoted the proliferation of beneficial gut microbes such as *Bifidobacterium*. A meta-analysis incorporating nine studies with a total of 303 participants assessed the therapeutic effects of FMT on obesity and/or metabolic syndrome. The analysis demonstrated that, within six weeks following FMT treatment, participants exhibited lower fasting blood glucose, HbA1c, and insulin levels compared to the placebo group, along with higher HDL levels. However, no significant differences were observed between groups in other metrics, such as body weight, BMI, HOMA-IR, and total cholesterol [122].

In small-scale clinical trials involving patients with NAFLD, FMT was not superior to probiotic preparations in improving blood lipid profiles or liver function. However, FMT demonstrated favorable effects on hepatic fat attenuation (quantified via FibroScan) [123]. Current explorations into FMT have introduced novel approaches, such as the use of washed microbiota transplantation (WMT) as an alternative to conventional FMT. Studies demonstrate that WMT effectively reduces serum uric acid levels and improves gout symptoms in patients with acute or recurrent gout [124].

Research on FMT has made significant advancements; however, challenges remain in reducing recipient risks, determining the optimal transplantation dosage, and establishing standardized donor selection and screening criteria. Furthermore, most published clinical studies to date have short follow-up durations, hindering comprehensive assessment of long-term effects and safety concerns [96].

##### Drug Modulation of Gut Microbiota

Certain drugs, such as metformin, can treat T2DM by modulating the composition of the gut microbiota in patients with T2DM. The baseline characteristics of the gut microbiota influence the therapeutic efficacy of metformin [125]. Metformin treatment can increase the abundance of *Akkermansia muciniphila* (a mucin-degrading bacterium) and enhance the population of SCFA-producing bacteria in the gut of patients with T2DM [126].

In recent years, various GLP-1 agonists, such as liraglutide, dulaglutide, and semaglutide, have been used to treat T2DM and improve obesity [6]. Additionally, some medications, such as somatostatin, lower the chance of clinical mortality from cardiovascular and renal reasons [127]. Some other drugs, such as dapagliflozin and enagliflozin, are SGLT2 inhibitors. These drugs inhibit renal tubular glucose reabsorption and promote urinary glucose excretion, achieving glycemic control via direct renal effects. Additionally, they can reduce glomerular hyperfiltration in patients with T2DM and help restore kidney function [7].

Smits et al. [128] found that liraglutide and sitagliptin had no significant impact on gut microbiota composition. Shung et al. [129] found that empagliflozin demonstrated beneficial effects on microbiota in patients with T2DM in terms of improving the colonies of *Eubacterium*, *Roseburia*, and *Faecalibacterium* and *Firmicutes* and *Bacteroidetes*, respectively, to the detriment of *Escherichia-Shigella*, *Bilophila*, *Hungatella*, and *Acinetobacter baumannii* in short-duration studies.

Table 1 consolidates gut microbiota changes, pathogenic mechanisms, and therapeutic interventions related to metabolic diseases (obesity, T2DM, NAFLD, hyperlipidemia) covered in this review.

## 4. Traditional Chinese Medicine (TCM)

### 4.1. TCM for Metabolic Diseases

Gut microbiota is closely related to the development of metabolic diseases. TCM can enhance the proliferation of beneficial bacteria and inhibit the growth of conditionally pathogenic bacteria by remodeling the structure and function of the gut microbiota, thereby improving the metabolic homeostasis of the host and providing novel interventions for the prevention and treatment of metabolic diseases. TCM contains various bioactive components, such as polysaccharides [159], flavonoids, and alkaloids [160], which might improve insulin resistance and reduce blood glucose and lipid levels, hence exerting therapeutic effects on metabolic disorders. Chronic inflammation and immune dysregulation are frequently associated with metabolic diseases. The anti-inflammatory and immunomodulatory effects of TCM help to reduce the inflammatory response, restore immune function and improve the disease state.

#### 4.1.1. TCM for T2DM

TCM holds that diabetes belongs to “Xiao Ke disease”. Medical professionals have long promoted the treatment of “Xiao Ke disease” based on the following principles: tonifying qi and nourishing Yin; clearing heat; encouraging the production of bodily fluids; relieving thirst; and nourishing the liver and kidneys. TCM, when lowering blood sugar, takes a holistic approach, treats based on syndrome differentiation, and follows the principle of “preventing diseases before they occur and preventing the progression of existing diseases”. It correctly evaluates the state of diabetes and applies focused, efficient interventions [161].

Single TCM exert hypoglycemic effects through multiple pathways. Single TCMs such as witch hazel, *Astragalus membranaceus* [162] and *Pueraria lobata* [163] have been proven effective. Polysaccharide constituents in TCM can ameliorate T2DM symptoms by modulating the insulin signaling system [164]. *Lycium barbarum* polysaccharides can enhance GLP-1 expression and modulate glucose metabolism in diabetic mice via activating the IRS/PI3K/Akt signaling pathway and influencing gut microbiota [165]. TCM can also treat T2DM through the regulation of microbiota. Research indicates that berberine can markedly enhance the prevalence of probiotics like *Bifidobacterium* and *Lactobacillus acidophilus*, while suppressing harmful bacteria such as *Proteobacteria*, and mitigating inflammatory reactions by optimizing gut microecology [166,167].

Moreover, TCM compound prescriptions exhibit features of many components, multiple targets, and several systems of action, resulting in a relatively intricate mechanism of action. The traditional Chinese medicinal therapy aimed at fortifying the spleen, dispelling turbidity, nourishing Yin, alleviating heat, and enhancing bodily fluid production is essential in the prevention of diabetes and its consequences. Liuwei Dihuang Wan has the effect of nourishing kidney yin and is widely used in the treatment of T2DM and its complications [168]. Compound Pearl Lipotropic Capsules can improve cardiac function in T2DM mice by lowering fasting blood glucose, inhibiting weight loss, and alleviating lipid metabolism disorders [169]. Other studies have shown that Compound Pearl Lipotropic Capsules can also improve coronary atherosclerosis by regulating inflammation and attenuating cell apoptosis [170]. Moreover, TCM compound prescriptions such as Erchen Decoction can inhibit lipolysis by regulating the IRS1/AKT/PKA/HSL signaling pathway, regulate the content of propionic acid, a metabolic product of the intestinal flora, and improve obesity and insulin resistance.

#### 4.1.2. TCM for Obesity

According to TCM, the pathogenesis of obesity is closely related to dysfunction of the spleen and stomach, internal generation of phlegm and dampness, qi stagnation and blood stasis, deficiency and decline of kidney Yang, improper diet, lack of exercise and emotional factors. The main causes of obesity are splenic and stomach dysfunction, as well as internal phlegm and dampness production. Qi stagnation, blood stasis, and kidney Yang deficiency can all make obesity worse. To cure obesity, TCM promotes the body’s consumption of energy substances, reverses the imbalance in gut microbiota brought on by obesity, and inhibits the digestion, absorption, synthesis, and inappropriate storage of fat. Research indicates that cassia seeds enhance liver lipid metabolism by activating genes associated with fatty acid oxidation, such as PPARα, consequently influencing appetite regulation [171]. Lotus leaf extract significantly mitigates obesity by diminishing digestive capacity, lowering lipid and carbohydrate absorption, and improving lipid metabolism regulation and energy expenditure [172]. Furthermore, *Caulis spatholobi* has demonstrated a substantial impact on diet-induced obesity in mice. It can impede weight gain, diminish obesity, sustain glucose homeostasis, decrease insulin resistance, and slow down hepatic steatosis by elevating the expression levels of genes associated with the activation and thermogenesis of brown adipose tissue in obese mice [173]. Poria cocos polysaccharides can alleviate obesity by regulating the gut microbiota–SCFAs-FGF21/PI3K/AKT signaling pathway, enhancing intestinal barrier function, and improving insulin resistance in adipose tissue [174]. They diminish fat accumulation and regulate weight by modulating fat metabolism, curbing hunger, enhancing metabolism, and elevating energy expenditure.

#### 4.1.3. TCM for Hyperlipidemia

In recent years, TCM has received increasing attention in the treatment of NAFLD due to its multi-target and multi-channel mechanism of action [175], such as *Salvia miltiorrhiza*, *bupleurum*, *yinchen*, *panax notoginseng*, *Gynostemma pentaphyllum*, etc., which can reduce blood lipids and improve blood circulation.

Single prescriptions in TCM, including *Salvia miltiorrhiza*, *Polygonum multiflorum*, *Artemisia argyi*, *Notoginseng*, and *Gynostemma pentaphyllum*, exhibit properties that reduce blood lipids and enhance blood circulation. Salvia *miltiorrhiza* promotes blood circulation, alleviates blood stasis, cools the blood, eliminates abscesses, nourishes the blood, and calms the mind. Research indicates that *Salvia miltiorrhiza* may provide a therapeutic effect on NAFLD by regulating lipid metabolism, enhancing microcirculation, and suppressing inflammatory responses. Research indicates that *Salvia miltiorrhiza* can mitigate hepatic inflammation, steatosis, and fibrous tissue formation in mice with NAFLD [176]. *Bupleurum* possesses properties that calm the liver, alleviate depression, harmonize internal and external elements, and elevate Yang qi. It can enhance the microcirculation of the liver and facilitate the repair and regeneration of hepatic cells. Research indicates that saikosaponin can modulate lipid metabolism by suppressing the expression of genes that hinder fatty acid biosynthesis and by stimulating the expression of genes that facilitate fatty acid breakdown [177]. *Astragalus membranaceus* has the functions of tonifying qi and raising Yang, benefiting the defense and consolidating the exterior, promoting diuresis and reducing swelling, and supporting toxins and promoting muscle growth. Studies have shown that *Astragalus membranaceus* activates the AMPK/PPAR-α pathway to inhibit the key factor SREBP-1 of lipid synthesis, reduces lipid accumulation in the liver, and at the same time regulates the balance of gut microbiota, reduces the Firmicutes/Bacteroidetes (F/B) ratio, and restores the homeostasis of intestinal microecology [178]. Liu et al. [14] investigated the effects of astragaloside, the active component of *Astragali*, a TCM, on rats with NAFLD. The results indicated that it effectively mitigated metabolic disorders induced by high-fat diets, including a reduction in hepatic fat degeneration, inhibition of mass increase, and enhancement of insulin resistance.

Compound formulas, such as Xuezhikang and Xuefu Zhuyu Tang, can be used to regulate blood lipids, prevent atherosclerosis and reduce hepatic fat deposition and inflammation through anti-inflammatory, antioxidant, and liver function improvement mechanisms.

#### 4.1.4. TCM for Hyperuricemia and Gout

Gout is induced by the accumulation of monosodium urate crystals in the synovial fluid and other body tissues of patients with hyperuricemia. Its attacks typically induce inflammation and erythema in the joints, thus causing damage to the joints [179]. Some medicines, such as *Rhizoma Coptidis*, *Eucommiae Cortex* [180], Huanglaguo [181], and Sanmiao Wan [182], have shown efficacy in enhancing gout-induced pathological damage and reducing inflammatory responses. Berberine, the active ingredient in Rhizoma Coptidis, can ameliorate the inflammatory response injury in the joints and their surrounding tissues by decreasing the level of inflammatory factors, regulating the gut microbiota, and promoting the excretion of uric acid [183]. Compound formulas such as Gout Drink and Si Miao Pill are used to relieve gout symptoms and reduce uric acid levels. Similarly, some drugs have combined therapeutic effects. For example, gout patients and diabetic patients often coexist [184]. Some herbs can simultaneously reduce blood glucose levels and treat gout. Polysaccharides extracted from *Dioscorea* can regulate the expression of urate transporter proteins in hyperuricemic mice by inhibiting xanthine oxidase to lower serum uric acid [185]. Meanwhile, diosgenin can mitigate diabetic complications by regulating glucose–lipid metabolism, protecting pancreatic β-cells, improving insulin resistance, and inhibiting oxidative stress and inflammatory responses [186]. TCM has some therapeutic effects on hyperuricemia and gout in clinical settings, but there are drawbacks. individualized care focused on the diagnosis and treatment of each syndrome is emphasized in TCM. Achieving a single treatment standard is challenging, though, because patients differ greatly from one another and treatment outcomes can differ as well. Furthermore, certain Chinese patent medications and traditional Chinese remedies have harmful side effects. They must be used appropriately and consistently since they have the potential to impair the functions of other organs while they are being treated.

The functions of some TCMs are shown in Table 2.

### 4.2. Extraction of Active Ingredients of TCM

The first key step in the functioning of TCM lies in the extraction of its active ingredients; the basic principle involves transferring the active constituents of the herbs into the solvent to create a TCM extract. The active ingredients of herbs typically comprise various compounds with different solubilities. By choosing suitable solvents and extraction conditions, the target components can be effectively extracted from the herbs. Traditional methods of TCM extraction include decoction, maceration, refluxing, percolation, and fermentation. These methods play a fundamental role in the pharmaceutical process of TCM, but there are some drawbacks, such as high loss of active ingredients, organic solvents that may interact with the active ingredients during the extraction process, and non-active ingredients that cannot be removed to the maximum extent.

With the rapid development of science and technology, modern extraction technology has become a research hotspot with significant advantages. They mainly include supercritical fluid extraction (SFE), membrane separation technology method, microwave extraction technology, semi-mimetic extraction technology, dynamic countercurrent extraction technology, enzyme-engineered extraction method, ultrahigh-pressure extraction technology, ultrasonic-assisted extraction technology, etc. [15].

These technologies markedly improve extraction efficiency and purity by accurately isolating certain bioactive components, including flavonoids, alkaloids, and polysaccharides. SFE demonstrates significant advantages in isolating active chemicals from natural plants, particularly with a high selectivity for alkaline compounds [195]. Silva et al. [196] utilized ultrasound-assisted technology to extract bioactive components from *ciriguela* peels, resulting in superior extraction rates and antioxidant activity compared to conventional and microwave extraction methods. These studies demonstrate that contemporary extraction methods can boost both extraction efficiency and purity, as well as augment the biological activity of substances.

Selecting the right extraction technique is crucial for studies on fermented TCM. Probiotic components’ fermentability and bioavailability will be impacted by various extraction methods. For example, flavonoids can be effectively extracted using ultrasonic extraction technology. During fermentation, these flavonoids can be converted into molecules with increased anti-inflammatory and antioxidant properties, increasing their positive effects on the gut microbiota. The extraction effect of the active ingredients in TCM can be maximized by carefully selecting the extraction technique. This will increase the probiotics’ ability to use these ingredients during fermentation and, ultimately, the positive effects of TCM fermentation products on intestinal function [197].

### 4.3. TCM Fermentation Methods

#### 4.3.1. Traditional Fermentation

As modern Chinese medicine progresses, the fermentation of TCM has increasingly garnered favor within the medical community. To increase the variety of medicines and adapt to clinical requirements.

TCM uses microorganisms to change their original properties, improve existing ones, or create new ones under specific environmental conditions (such as temperature, humidity, etc.). TCM can develop new effects or improve its original qualities through fermentation [198]. TCM fermentation methods can be divided into two categories: modern Chinese medicine fermentation methods and traditional herbal fermentation methods [198]. The majority of conventional fermentation methods utilize natural fermentation, also known as solid fermentation, and are performed in aerobic conditions. The herbs experience a biological transformation through the action of bacteria and enzymes under specific environmental conditions, resulting in fermentation or the creation of a moldy coating. In TCM, red yeast rice and Liu Shen Qu are prevalent TCM products generated via natural fermentation processes, including solid-state fermentation. Red yeast rice is generated through the fermentation of rice and red yeast. Research indicates that red yeast rice can lower blood triglyceride levels by blocking fatty acid production and enhancing fatty acid β-oxidation. The active constituents, including lovastatin and monene chemicals, can enhance insulin production from pancreatic β-cells and elevate insulin levels, thus addressing metabolic disorders such as hyperlipidemia and diabetes. The dosage of red yeast rice must be meticulously regulated to minimize undue injury to the human body [199].

The fermentation forms utilized in modern fermentation technology can be categorized into solid-state fermentation, liquid fermentation, and dual-phase fermentation technology. There are clear differences between solid-state fermentation and liquid-state fermentation in terms of their microbial biological activity and metabolic spectrum. The solid-state fermentation process retains its traditional process characteristics and can retain and promote the growth and metabolism of target microorganisms. The liquid fermentation process better meets the requirements of industrial production, and its parametric regulation accuracy can greatly improve the stability of product quality and batch-to-batch consistency.

Traditional extraction methods are characterized by complex procedures, coarse processing techniques, low levels of equipment automation, insufficient product purity, potential residues of toxic and harmful organic solvents, and outdated quality control measures [200]. Studies have demonstrated that the fermentation of traditional Chinese medicine can enhance the content of active compounds in medicinal materials, thereby improving therapeutic efficacy. Moreover, the fermentation process may generate novel bioactive components, transform or degrade toxic substances, and increase the extraction efficiency of active ingredients [201]. Research progress on TCM fermentation and the profile of active substances derived.

#### 4.3.2. Probiotic Fermentation

Probiotic fermentation of herbal medicines enhances their biological activity and produces beneficial compounds by decomposing or converting substrates into new active components with the aid of enzymes. Probiotic-fermented TCM is more bioactive, less toxic, and has fewer side effects than TCM fermentation, according to the microbial hypothesis. Moreover, the quality and safety of the fermented products are enhanced, and the fermentation process is more easily controlled [11]. Probiotic strains must be strictly screened and must possess specific health-promoting characteristics, such as resistance to gastric acid and bile, the ability to adhere to intestinal epithelial cells, antimicrobial activity, etc. At the same time, it is necessary to ensure that they pose no harm to the host and eliminate any strains that may be harmful to the host. The production of probiotics should be carried out in an environment that complies with Good Manufacturing Practice (GMP) for pharmaceuticals or Good Food Production practices, ensuring hygiene conditions during the production process and reducing the risk of contamination. The equipment should be maintained, cleaned and verified regularly to ensure its normal operation and the stability of product quality. A series of safety assessments need to be conducted after production [202].

##### Enhancing Active Ingredients

Herbal fermentation can enhance the bioactivity of herbal medicines. Compared with traditional herbal processing methods, the fermentation of herbal medicines utilizing probiotics can enhance the bioactivity of herbal medicines under mild processing conditions and promote the release of active ingredients, such as organic acids and polysaccharides, from herbal medicines. The fermentation products can enhance the immunity, antioxidant properties and disease resistance of the host [12,203]. Fermented herbs with probiotics exhibit a synergistic impact resulting from the interaction between the herbs and the probiotics. Probiotics generate extracellular enzymes (e.g., pectinase, cellulase, and protease) during fermentation, which can hydrolyze the cell walls of herbal medicines and optimize the release of medicinal compounds. The proteins, vitamins, and trace elements present in TCM enhance the growth, reproduction, and metabolism of probiotics [198]. Wei et al. [204] fermented Ganmai Dazao Decpotion with lactic acid bacteria. The results showed that the content of gamma-aminobutyric acid after fermentation increased by 12.06% compared with that before fermentation, and the content of small molecule substances related to antioxidant activity increased significantly. Yong et al. [205] discovered that fermenting *turmeric* with *Lactobacillus* fermentum could enhance the curcumin content in turmeric by 9.76%. This may result from the enzymatic conversion of *Lactobacillus* fermentum during fermentation, which demethylates curcumin. Macromolecular substances can be transformed into more absorbable forms via glycosylation and deglycosylation processes.

Guo et al. [12] discovered that co-fermentation of Angelica sinensis tonic soup with the probiotic *Lactobacillus plantarum* showed increased activity in inhibiting α-glucosidase, inhibiting DPPH radical scavenging, and anti-glycosylation relative to the original herbal product. The β-glucosidase generated during the fermentation of *Lactobacillus plantarum* catalyzes the hydrolysis of flavonoid glycosides in Danggui Buxue Decoction into their aglycone forms (calycosin and formononetin). Aglycone exhibits superior intestinal absorption and bioavailability.

##### Reducing the Toxicity

The decomposition and transformation of microorganisms can diminish the toxicity of certain herbs by degrading or changing the structure of their toxic constituents, changing the medicinal properties, reducing toxic side effects and expanding the clinical indications of herbs. The fermentation process diminishes the toxicity of specific herbs by enzymatic hydrolysis, adsorption, metabolic transformation, and antagonistic actions [201].

Research indicates that fermentation can significantly diminish the levels of highly toxic alkaloids, including aconitine and methaconitine, in Huafeng Danwanmu, while simultaneously enhancing the quantities of aconitine and benzoyl ketone in the advantageous compound benzoyl. This results from the diverse enzymes generated by microbes during fermentation, including the cytochrome P450 enzyme, among others. The modification of harmful alkaloid structures can diminish medication toxicity [10].

##### Generating New Compounds

Moreover, probiotic fermentation can transform herbal constituents into novel bioactive molecules, thereby imparting new pharmacological capabilities to herbal medications. Okamoto et al. [206] used *Lactobacillus plantarum* SN13T to ferment Artemisia quinquefolium extracts. The findings indicated a significant inhibition of IL-8 release from HuH-7 cells compared to unfermented extracts. Additionally, two novel bioactive compounds, catechol and short-tongue *pikinolide*, were identified. Catechol has substantial antioxidant and anti-inflammatory effects and can treat diabetes through numerous pathways such as decreasing insulin resistance, reducing oxidative stress, and modulating mitochondrial function [207]. Consequently, the fermentation of herbs utilizing *Lactobacillus plantarum* SN13T is a crucial method for acquiring active compounds with therapeutic potential.

Probiotic fermentation of herbs not only increases the amount of bioactive natural products but also improves the host’s gut microbiota and immune system. It can transfer the gut microbiota’s metabolic transformation of the active compounds in herbal medicines to in vitro during the manufacturing and preparation process, which has a wide range of possible applications.

#### 4.3.3. Fermentation Strains

*Lactobacillus* and *Bifidobacteria* are the two main types of fermenting bacteria that are commonly used in the probiotic fermentation of TCM. *Lactobacillus* fermentation has been used in TCM such as *dendrobium ironum* [208], *red ginseng* [209], *licorice* [210], and *Astragalus membranaceus* [211]. *Lactobacillus* can produce lactic acid and reduce pH during the fermentation process, which helps to maintain the stability of the components of TCM and improve the conversion of active ingredients in TCM. Some TCMs also enhance the antioxidant capacity after fermentation [209]. For instance, utilizing *Bifidobacteria* to ferment *Pueraria mirifica* [212] can increase the effectiveness and therapeutic effects. When using probiotics in fermented products, several factors must be taken into account, especially their viability and their large presence when consumed. Studies show that lactic acid bacteria have a high tolerance to very low-pH conditions and can survive in the acidic conditions of the stomach. Some lactic acid bacteria strains have multiple drug resistance, indicating that they can survive certain drug treatments and may help maintain the gut microbiota and restore the microbial community during antibiotic treatment [11,213,214].

Others, such as *Saccharomyces cerevisiae*, a typical species of the fungus, thrive in an acidic and high-sugar environment, making them ideal for the fermentation of TCM ingredients that contain a lot of sugar. For instance, the active components in *ginseng* might dissolve more quickly when fermented with yeast [201]. Some fungi, particularly medicinal fungi, have been used to ferment herbal or TCM preparations. Numerous chemical processes, including glycosylation, hydroxylation, methylation, and others, can be triggered by fungal fermentation, which also yields many secondary metabolites. *Aspergillus niger*, *Aspergillus erythropolis*, and other fungus are commonly utilized. Moon et al. [215] fermented the roots of *Salvia miltiorrhiza* with *Aspergillus oryzium*, increasing the total phenol content and flavonoid content by 1.3 times and 1.2 times, respectively, enhancing the antioxidant potential. The originally hydrophobic active components (such as phenolic acids and tanshinone) were transformed into more hydrophilic derivatives (such as esterified fatty acids) through fermentation, improving the solubility in the aqueous phase and enhancing the bioavailability. Some other combined probiotic or microbial fermentations offer new possibilities for improving and increasing microbial fermentation. These fermenting bacteria are used in herbal fermentation with the goals of increasing the release and bioavailability of active components, decreasing toxicity, improving efficacy, and creating novel bioactive compounds. Thus, choosing the right strains of fermenting bacteria is crucial to achieving particular fermentation objectives. By comparing the characteristics of different strains and combining their interactions with the components of TCM, the fermentation process can be better optimized, and the quality and efficacy of TCM can be improved.

### 4.4. Mechanisms of TCM in Treating Metabolic Diseases

The mechanism of TCM in treating metabolic diseases, especially through regulating gut microbiota, has become a popular research topic.

#### 4.4.1. TCM Functions as a Natural Prebiotic

TCM contains a variety of biologically active ingredients, which can affect the gut microbiota through multiple pathways and play an important role in restoring bacterial homeostasis due to their ability to multi-target and comprehensively control the gut microbiota [9]. The active ingredients in TCM can change the composition of gut microbiota, such as polysaccharides, flavonoids, saponins and other components, and they can act as prebiotics to promote the growth of probiotic bacteria like *Bifidobacteria* and *Lactobacilli* while also preventing the reproduction of harmful bacteria such as *Enterobacteriaceae*, thus improving the structure and function of gut microbiota. Studies have shown that long-term consumption of *ginseng* can effectively increase the diversity and abundance of gut microbiota, thereby increasing the number of *Bifidobacteria* and *Lactobacillus*, and improving the health level of the host [216]. Fang et al. [217] investigated the gut microbiota of diabetic mice by orally administering Dendrobium polysaccharides, discovering a reduction in the ratio of the *Firmicutes* phylum to the *Bacteroidetes phylum*, alongside an increase in the abundance of beneficial gut bacteria, specifically *Lactobacillus*, *Bifidobacterium*, and *Actinobacteria*.

TCM and its bioactive constituents exhibit significant promise in modulating gut microbiota; nevertheless, the dose-dependent effects and clinical consequences require further elucidation through more high-quality research. Recent studies indicate a dose-effect relationship between the regulatory impact of TCM compound prescriptions and their components on intestinal flora, as well as clinical efficacy within a certain dosage range. A systematic review encompassing 13 randomized controlled studies indicates that the efficacy of TCM compound prescriptions in enhancing blood glucose levels and modulating gut microbiota in the management of T2DM is dose-dependent. The precise dose–response relationship requires further validation through more rigorous investigations [218].

The human microbiome exhibits considerable inter-individual variability, potentially influencing the regulatory impact of TCM on gut microbiota and clinical outcomes. The therapeutic efficacy of TCM in modulating intestinal flora may be constrained by factors such as the drug resistance of the flora and individual variations in the constituents of the medicine. Currently, the majority of research concentrates on animal models and in vitro studies, but human trials are rather limited. Future research should investigate the mechanisms of action of TCM within the human body, elucidate its dose-dependent effects and clinical outcomes, and consider the influence of human microbiome heterogeneity on therapeutic efficacy, to enhance the treatment strategies of TCM.

#### 4.4.2. TCM Functions by Improving the Intestinal Mucosal Barrier

The intestinal mucosal epithelial cells and the tight junctions between the cells jointly constitute the intestinal mucosal mechanical barrier [219]. Patients with metabolic diseases frequently exhibit intestinal barrier impairment, leading to bacterial and endotoxin translocation and triggering inflammatory reactions. A clinical trial investigated the impact of duodenal mucosal remodeling (DMR) in conjunction with GLP-1RA on T2DM. Following three months of treatment, the glycated hemoglobin (HbA1c) levels in patients exhibited a negative correlation with the α-diversity of the intestinal microbiota, whereas alterations in the liver proton density fat fraction (PDFF) were strongly connected with the β-diversity of the intestinal microbiota. This suggests a correlation between alterations in gut microbiota diversity and metabolic enhancement [220]. TCM can strengthen the tight connections of intestinal epithelial cells, diminish the secretion of inflammatory mediators, and improve the structure of the microbiota, hence enhancing intestinal barrier function. At the molecular mechanism level, TCM achieves this function by regulating specific tight junction proteins (such as ZO-1 and occludin). These proteins are crucial for maintaining the integrity of the intestinal mucosa.

The plant ingredient resveratrol exerts a beneficial regulatory influence on lipid and glucose metabolism. Resveratrol is a naturally occurring polyphenol present in various plants and can mitigate diabetes and hepatic steatosis [221]. Zhang et al. [187] found that *resveratrol* and its metabolites (such as resveratrol 3-O-sulfate) can upregulate the mRNA expression levels of tight junction proteins (such as ZO-1, occludin, claudin-1, etc.) in intestinal epithelial cells, thereby enhancing the integrity of the intestinal mucosal barrier and reducing intestinal permeability. Furthermore, resveratrol can control the diversity and composition of gut microbiota as well as support intestinal flora homeostasis. While preventing the growth of dangerous bacteria, it can boost the number of good bacteria (such as *Lactobacillus* and *Bifidobacterium*). Enhancement of this microbiota balance lessens the harm caused by inflammatory reactions to the intestinal mucosal barrier and contributes to the stability of the intestinal milieu.

Duan et al. [222] demonstrated that Poria cocos polysaccharides can modify the microbial composition of the intestinal tract, thereby enhancing intestinal barrier function, significantly improving the physiological condition of the intestinal tract in mice, and elevating the concentration of SCFA in the small intestinal contents. Polysaccharides from *Atractylodes macrocephala* have been shown to regulate the SCFA metabolic products of the gut microbiota. SCFAs enhance intestinal barrier function in a number of ways, including by regulating the expression of tight junction proteins, maintaining the integrity of intestinal epithelial cells, and increasing the expression of mucins. *Atractylodes macrocephala* polysaccharides can control the amounts of inflammatory factors, including TNF-α, IL-1β, and IL-6. *Atractylodes macrocephala* polysaccharides can reduce intestinal inflammation and aid in the restoration of the intestinal mucosal barrier by controlling these inflammatory factors [223].

#### 4.4.3. TCM Functions by Regulating Gut Microbiota

Studies have shown that metabolic diseases such as obesity, T2DM, and NAFLD are closely related to the structural and functional disorders of gut microbiota. An imbalance of gut microbiota is regarded as a characteristic of metabolic diseases [74,224,225].

Several studies have shown that TCM and its combinations can enhance glucose and lipid metabolism by controlling gut microbiota homeostasis, improving the intestinal mucosal barrier, or modulating the release of endogenous functional molecules [226]. The modulation of gut microbiota by TCM affects glycolipid metabolism mainly by regulating flora, such as mucin-degrading flora, e.g., *Akkermansia muciniphila*, beneficial flora with anti-inflammatory effects, e.g., *Faecalibacterium prausnitzii* and *Roseburia* spp., the abundance of LPS- and SCFA-producing flora such as *Butyrivibrio*, *Bifidobacterium* and *Megasphaera*, as well as the abundance of bile salt hydrolase (BSH)-carrying flora (e.g., *Lactobacillus*, *Bifidobacterium*, etc.), to improve intestinal mucosal integrity, protect intestinal barrier function, and improve endotoxemia and inflammatory response.

TCM, such as *Huanglian* and *Astragalus*, can improve glucose metabolism and insulin resistance in T2DM patients by regulating gut microbiota. For example, berberine in Huanglian can increase the content of *Akkermansia muciniphila* in the intestine and improve insulin sensitivity [13]. In addition, Chinese herbs combat obesity by regulating gut microbiota and reducing energy intake and fat accumulation. Studies have demonstrated that lotus leaf extract and its bioactive compounds can mitigate diet-induced obesity through the activation of genes associated with brown adipose tissue, thereby improving glucose tolerance and insulin sensitivity [188].

TCM can treat NAFLD by improving the gut microbiota and reducing hepatic fat deposition and inflammatory responses. Tanshinones derived from *Salvia miltiorrhiza* can diminish the population of ethanol-producing bacteria in the gastrointestinal tract and alleviate the liver’s burden.

For example, tanshinones from *Salvia miltiorrhiza* can reduce the number of ethanol-producing bacteria in the gut and reduce the burden on the liver [91]. Similarly, some herbal compounds can be used to treat metabolic diseases. For example, the main hypoglycemic active ingredients of Artemisia Incarnata Tang are gardenia glycosides, rhubarb phenol, and rhubarb acid. Zhao et al. [193] studied the therapeutic effect of Yinchenhao Decoction on mice with acute pancreatitis. The results showed that Yinchenhao Decoction could increase the diversity of the intestinal flora in mice and regulate the flora structure to a normal level, thereby affecting the synthesis, binding, and excretion of bile acids, reducing the accumulation of toxic bile acids in the liver, and demonstrating potential therapeutic effects on NAFLD. There is also Ge Gen Baicalin Lian Tang, which consists of Ge Gen, *Scutellaria baicalensis*, *Rhizoma Coptidis*, and *Glycyrrhiza glabra*. By treating high-fat-diet-induced T2DM mice with Gegen Qinlian Decoction, Xu et al. [194] showed significant anti-hyperlipidemic effects, as well as modulation of the gut microbiota, increasing the abundance of beneficial bacteria, such as *Coprococcus*, *Bifidobacterium*, *Blautia* and *Akkermansia*. And metabolomics analysis revealed that Gegen Qinlian Decoction treatment elevated the level of specific bile acids, which may improve diabetic symptoms by regulating lipid metabolism.

In summary, this article systematically reviews the research progress of TCM in targeted treatment of metabolic diseases through regulation of the gut microbiota, with a particular emphasis on elucidating the core role and mechanism of probiotic fermentation technology in enhancing the therapeutic efficacy of TCM. Metabolic diseases, including obesity, T2DM, NAFLD, and gout, are closely associated with gut microbiota imbalance. TCM demonstrates unique therapeutic potential by regulating microbial balance and host metabolism via multiple targets. The primary mechanism involves probiotics enhancing active components through fermentation, releasing bioactive substances, breaking down the cell walls of TCM to release active components such as polysaccharides and flavonoids, and improving their bioavailability. Probiotic fermentation regulates the structure of the gut microbiota, promotes the proliferation of beneficial bacteria, and inhibits pathogenic bacteria (e.g., *Enterobacteriaceae*). Restoring microbial diversity alleviates the dysbiosis characteristics associated with metabolic diseases. Enhancing the intestinal barrier and immune regulation repairs the intestinal mucosal barrier: fermentation products alleviate chronic inflammation by upregulating the expression of tight junction proteins (ZO-1, occludin), reducing intestinal permeability, and inhibiting endotoxin (LPS) translocation. Regulating immune responses: fermented TCM (e.g., tanshinone) suppresses pro-inflammatory factors (TNF-α, IL-17), promotes the secretion of anti-inflammatory factors (IL-10), and mitigates inflammatory damage caused by metabolic diseases. Regulating the production of SCFAs: fermented TCM promotes the proliferation of SCFA-producing bacteria (e.g., *Roseburia*), increases metabolites such as butyric acid and propionic acid, and improves insulin resistance and lipid metabolism. Probiotic-fermented TCM offers a natural and safe therapeutic strategy for metabolic diseases through a multidimensional mechanism of “microbiota remodeling—metabolic regulation—immune balance”. This approach represents an innovative integration of traditional medical wisdom with modern biotechnology (Figure 2).

## 5. Challenges and Future Directions

Although progress has been made in the research on the pathogenesis and treatment strategies of metabolic diseases, key challenges still exist. The complexity of the gut microbiota-host interaction constitutes a core obstacle. The dynamic interaction between microbial metabolites (such as SCAFs, LPS) and host metabolic pathways involves a multi-level regulatory network, and its species-specific effects and spatiotemporal dynamic characteristics have not been fully clarified.

Furthermore, the standardization of TCM practice is hindered by differences in extraction methods, inconsistent quality control, and limited mechanism validation. Although modern techniques such as SFE can enhance the yield of bioactive compounds, their scalability and cost-effectiveness still need further optimization. Emerging therapies like FMT and probiotic formulations show promise in preclinical studies. Still, their long-term safety, the possibility of dysbiosis recurrence, and efficacy in different populations remain to be investigated. Additionally, the multifactorial nature of metabolic diseases, influenced by genetic, epigenetic, and environmental heterogeneity, complicates the development of universal treatment strategies, necessitating a shift towards personalized approaches to account for individual differences in gut microbiota composition and host reactivity. Future research should prioritize integrating multi-omics technologies (metagenomics, metabolomics, proteomics) to comprehensively map the gut microbiota–host crosstalk and identify key microbial taxa and metabolites as therapeutic targets. For instance, the Global Microbiome Catalog (gcMeta) includes genomic information of microorganisms from various environments and research projects, supporting the archiving, standardization, analysis, retrieval, and visualization of microbiome data, and providing rich data support for biomedical research, including metagenomic studies. This has applications for treating metabolic diseases, such as microRNA at the genetic level in metabolic syndrome. In the field of molecular medicine, microRNA regulates metabolic syndrome at the genetic level. By controlling gene expression and inflammatory responses, it affects the transduction of insulin signals, insulin sensitivity, and metabolic pathways, thereby exerting effects on various metabolic diseases and metabolic syndrome.The treatment of metabolic syndrome with microRNA and the understanding of its molecular mechanism can provide new research directions and strategies [227]. Innovations in TCM modernization, such as probiotic fermentation (e.g., *Lactobacillus*-processed ginseng) and nanotechnology-based delivery systems, may enhance bioactive compounds’ stability and targeted specificity while reducing toxicity. By addressing these challenges through interdisciplinary collaboration and technological innovation, the field can transition from a broad therapeutic paradigm to precision medicine, ultimately improving treatment outcomes for patients with metabolic disorders.

## 6. Conclusions

This review systematically explains the synergistic effect of key probiotics (such as *Lactobacillus plantarum*, *Bifidobacterium*) and herbal active ingredients (berberine, tanshinone, polysaccharide), which can enhance the integrity of intestinal barrier. Regulation of microbial metabolites (e.g., SCFAs, bile acids); Suppress chronic inflammation. Fermentation technology not only improves the bioavailability of TCM ingredients, but also degrades toxic alkaloids (such as aconitine) and generates new anti-inflammatory and antioxidant metabolites. Based on the system regulation model of “microbiota-metabolism-immunity” axis, by integrating traditional herbal medicine experience with modern biotechnology, it can provide a new method for individualized precision treatment.

## Figures and Tables

**Figure 1 ijms-26-05486-f001:**
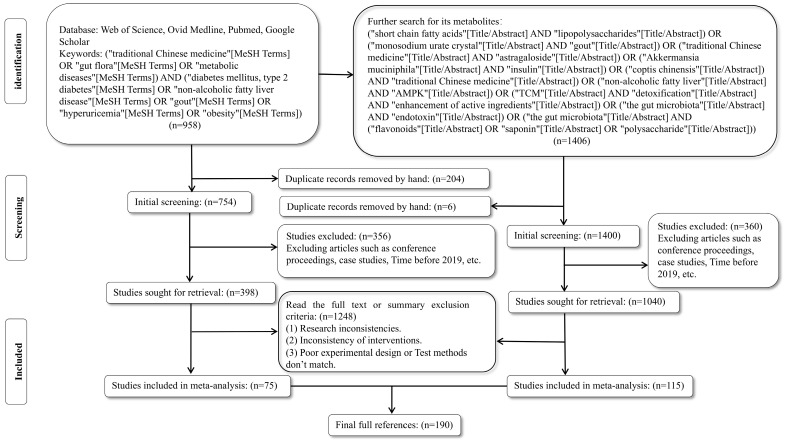
This flowchart shows the selection process of the studies included in this review.

**Figure 2 ijms-26-05486-f002:**
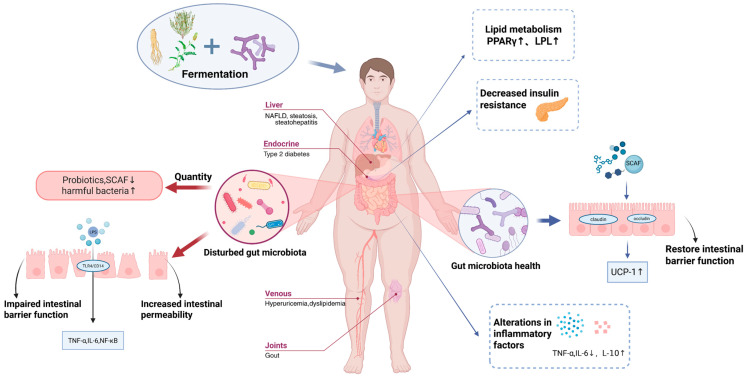
Probiotic-fermented TCM treats related metabolic diseases through the gut microbiota.

**Table 1 ijms-26-05486-t001:** Changes in the gut microbiota, critical mechanisms and intervention measures of the related metabolic diseases (obesity, T2DM, NAFLD and hyperlipidemia) are described above. The symbol “↑” denotes an increase in the relative levels of specific bacterial phyla within the gut microbiota associated with a particular disease, while “↓” signifies a decrease. The symbol “+” indicates that the therapeutic efficacy of certain interventions has been experimentally verified in human trials or animal studies, whereas “-” denotes interventions whose therapeutic effects have not yet been experimentally confirmed in human trials or animal studies.

**Metabolic Disease—Obesity, T2DM, NAFLD and Hyperlipidemia**							
**Metabolic Disease**	**Changes in Gut Microbiota** **(Increase/Decrease)**	**Critical Mechanisms**	**Intervention Treatment**		**Animal**	**Human**	**Ref.**
Obesity	*Firmicutes*↑, *Bacteroidetes*↓	-Pathogen: Increases energy absorption, SCFAs promote fat storage -Treatment: ① Inhibits BSH activity, increases the circulating levels of TDCA and TUDCA, inhibits intestinal carbonic anhydrase 1 expression, and reduces energy absorption. ② Modulates the composition of bile acids, inhibits the ileal FXR-FGF15 signaling pathway, promotes the FXR-SHP signaling pathway, and influences the browning of white adipose tissue.		Diet	Intermittent fasting	Ten- to eleven-hour time-restricted eating Fasting-mimicking diet	[116,130,131,132,133,134,135,136,137,138,139,140,141,142]
				Probiotics	*Lactiplantibacillus plantarum ZDY2013*	*Lactobacillus acidophilus*, *Lactobacillus casei*, *Lactobacillus rhamnosus*, *Lactobacillus bulgaricus*, *Bifidobacterium bifidum* and *Bifidobacterium longum*	
				Prebiotics and postbiotics	Non-digestible xylooligosaccharide Kudzu resistant starch	Butyrate and inulin anthocyanin and dietary fiber	
				FMT	+	A 90-day controlled open-label trial (16 patients) A meta-analysis incorporating 9 studies with a total of 303 participants	
**Metabolic Disease—T2DM**							
T2DM	*Bifidobacterium*↓, *Akkermansia muciniphila*↓	-Pathogen: Gut barrier damage, decreases insulin sensitivity -Treatment: ① SCFAs↑ ② Suppresses oxidative stress and intestinal inflammation, restores intestinal barrier integrity ③ Promotes glucose uptake and glycogen synthesis in IR-HepG2 cells to ameliorate insulin resistance ④ Modulates IRS-1/PI3K/AKT/Glut4 signaling transduction to improve the glucose sensitivity	**Intervention Treatment**	Diet	Time-restricted eating Ketogenic diet	Ten- to eleven-hour time-restricted eating Mediterranean diet Fasting-mimicking diet	[107,121,122,134,143,144,145,146,147]
				Probiotics	*Phascolarctobacterium faecium DSM 32890* *Lactobacillus paracasei subsp. paracasei NTU 101* *Lactobacillus reuteri J1*	*Lactobacillus. fermentum* strains K7-Lb1, K8-Lb1 and K11-Lb3+ *Lactobacillus. plantarum*	
				Prebiotics and postbiotics	Inulin Fructooligosaccharide Galactooli-gosaccharide *Lactobacillus plantarum* L-14 extracts (EPS)	Heat-killed *Akkermansia mucinophila*	
				FMT	+	During the recovery phase, utilizing feces collected during the weight loss phase for autologous FMT can help prevent weight regain.	
**Metabolic Disease—NAFLD**							
NAFLD	Ethanol-producing bacteria↑, *γ-Proteobacteria*↑	-Pathogen: Ethanol-induced liver injury, disorder of bile acid metabolism -Treatment: Restores the intestinal barrier integrity and function SCFAs↑, circulating leptin↓, inhibited FASN and ACC1	**Intervention Treatment**	Diet	+	Mediterranean diet	[106,109,123,148,149,150,151,152,153,154]
				Probiotics	*Desulfovibrio* vulgaris	High-concentration multi-strain probiotic formulation (including *Lactobacillus paracasei DSM 24733*, *Lactobacillus plantarum DSM 24730*, *Lactobacillus acidophilus DSM 24735*, *Lactobacillus delbrueckii subsp. bulgaricus DSM 24734*, *Bifidobacterium longum DSM 24736*, *Bifidobacterium infantis DSM 24737*, *Bifidobacterium breve DSM 24732*, and *Streptococcus thermophilus DSM 24731*+)	
				Prebiotics and postbiotics	Xylo-oligosaccharide Inulin Astragalus polysaccharides	Oligofructose	
				FMT	Gut microbiome of patients less prone to fatty liver caused by olanzapine exhibited an alleviation against fatty liver disease in rats.	FMT had a significantly higher healing efficacy on lean NAFLD	
**Metabolic Disease—Hyperlipidemia**							
Hyperuricemia	*Bacteroidetes*↑, Butyrate-producing bacteria↓	-Pathogen: Abnormal purine metabolism, increased uric acid synthesis -Treatment: ① Inhibits the activation of NLRP3 inflammasome and TLR4/MyD88/NF-κB signaling pathway ② ABCG2 in kidney and XOD in liver↑ ③ URAT1 and GLUT9 in kidney↓ ④ SCFAs↑, improves intestinal function	**Intervention Treatment**	Diet	Plant-based diet	-	[124,155,156,157,158]
				Probiotics	*Pediococcus acidilactici GQ01* *Akkermansia muciniphila* *L. rhamnosus R31, L. rhamnosus R28-1,* and *L. reuteri L20M3* *Limosilactobacillus reuteri HCS02-001*	*Lactobacillus gasseri PA-3 -*	
				Prebiotics and postbiotics	Polysaccharides Phenols Peptides Heat-killed *Pediococcus acidilactici* GQ01 *pasteurized Akkermansia muciniphila*	-	
				FMT	-	Washing microbiota transplantation	

**Table 2 ijms-26-05486-t002:** TCMs and compound prescriptions for treating metabolic diseases, and their main components and possible mechanisms.

TCM/Compound Prescription	Active Ingredient(s)	Mechanism of Action	Result (Experimental/Preclinical)	Ref.
Rhizoma coptidis	Berberine	Regulates gut microbiota, increases *Akkermansia muciniphila* abundance	Improves insulin resistance and lowers blood sugar	[13]
*Salvia miltiorrhiza*	Tanshinone I	Reduction of ethanol-producing bacteria reduces liver burden	Alleviation of NAFLD fatty liver degeneration	[91]
*Poria cocos*	β-glucan	Strengthens the gut barrier and promotes the added value of probiotics	Improvement of gut homeostasis in mice	[187]
Radix Astragali	Astragaloside A	Improves insulin sensitivity and alleviates liver steatosis	Improvement of T2DM and NAFLD	[162]
Lotus leaf	Ethanol extract	Regulates lipid metabolism by activating brown adipose tissue	Loses weight	[188,189]
*Dioscorea opposita*	Polysaccharide, saponin	Inhibits xanthine oxidase to reduce uric acid and regulates glycolipid metabolism	Reduces serum uric acid, improves renal function, and regulates blood glucose metabolism	[185]
Hawthorn leaf	Flavonoid	Regulates fat metabolism, suppresses appetite, and promotes energy expenditure	Alleviates obesity induced by a high-fat diet	[190]
*Tripterygium wilfordii*	Celastrol	Regulates energy intake and expenditure	Prevents energy excess caused by a high-fat diet	[191,192]
Liuwei Dihuang Pills	Polysaccharide, flavonoid	Regulates the balance of gut microbiota and enhances SCFA production	Improvement of diabetic complications	[168]
Fufang Zhenzhu Tiaozhi Capsule	Japanese honeysuckle	Lowering of fasting glucose, inhibition of lipid metabolism disorders, modulation of inflammation and apoptosis	Improves T2DM and alleviates coronary atherosclerosis	[169,170]
Yinchenhao Decoction	Gardenia glycosides, rhubarb phenol	Affecting the synthesis, binding and excretion of bile acids	Improves NAFLD	[193]
Gegen Qinlian Decoction	*Pueraria lobata*, *Scutellaria baicalensis*	Antilipidemic	Regulates lipid metabolism to improve diabetes symptoms	[194]

## Data Availability

Not applicable.

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
