# Peer review of "Gut Microbiota-Targeted Therapeutics for Metabolic Disorders: Mechanistic Insights into the Synergy of Probiotic-Fermented Herbal Bioactives"

_ijms, 2025, doi:10.3390/ijms26125486_

Round 1
Reviewer 1 Report
Comments and Suggestions for Authors
The review article "Gut Microbiota-Targeted Therapy for Metabolic Disorders: Synergistic Effects of Probiotic-Fermented Herbal Bioactives" presents interesting information related to therapies for metabolic disorders. Recommendations:
- The first chapter should be clearly designated as an Introduction. It must clearly present the aim of the study along with general background information on the topic.
- A more detailed discussion is recommended regarding the role of gut microbiota in digestive diseases. The following reference is suggested: https://doi.org/10.3390/jcm14082678.
- Given the vastness of the subject, it is advised to add a flow chart illustrating the selection process of the studies included in the review.
- Chapter 2 would benefit from a summary table that synthesizes the key findings of the analyzed articles.
- Table 1 is overly simplistic. It should either be expanded to include more relevant data or be removed if it does not add significant value.
- The topic of fecal microbiota transplantation is discussed only briefly. This subject deserves more in-depth analysis considering its therapeutic potential.
- Greater emphasis should be placed on probiotics, particularly by discussing studies that highlight their limitations in fully restoring the gut microbiota.
- In Chapter 4, it is recommended to address the importance of microRNAs in the context of metabolic syndrome. The following reference is useful: https://doi.org/10.3390/jcm14062054.
- A dedicated Conclusions chapter should be added to succinctly summarize the main findings and their clinical relevance.
Author Response
- 评论:第一章应明确指定为 Introduction。它必须清楚地介绍研究的目的以及有关该主题的一般背景信息。
响应:
我们非常感谢这个评论。根据这个建议,我们修改了我们的手稿,提供了更多细节,并提供了更多有用的知识。作为直接回应,我们修改了手稿的引言部分(第 1 页,第 39 行),以明确阐明研究目标,同时加强本研究的背景。具体来说,我们整合了与我们工作相关的其他基础知识,确保与当前的研究范式保持一致。
- 评论:建议更详细地讨论肠道微生物群在消化系统疾病中的作用。建议使用以下参考:https://doi.org/10.3390/jcm14082678。
响应:
我们衷心感谢审稿人的宝贵建议。
根据推荐的文献,我们在第 3.2.2.4 节(第 12 页,第 569 行)和第 3.2.1 节(第 8 页,第 347 行)中大大扩展了关于肠道微生物群在消化系统疾病中的作用的讨论。这些修订包括对疾病发病机制中微生物群失调的增强机制解释,并得到了引用参考文献的最新临床证据的支持。
- 评论:鉴于主题的广泛性,建议添加一个流程图,说明综述中纳入的研究的选择过程。
响应:
我们衷心感谢这个富有洞察力的建议。
为了响应您的建议,我们合并了一个全面的流程图(现在包含在第 3 页第 92 行的图 1 中),以系统地说明本综述的研究选择过程。该可视化示意图阐明了纳入/排除标准、筛选阶段和最终文章选择的基本原理,从而提高了方法学的透明度和可重复性。
- 评论:第 2 章将受益于一个汇总表,该表格综合了分析文章的主要发现。
响应:
我们衷心感谢审稿人的这个建设性建议。
作为直接响应,我们合并了一个全面的汇总表(现在包含在第 14 页第 625 行的表 1 中),该表综合了与肥胖、T2DM、高脂血症和 NAFLD 相关代谢紊乱相关的关键发现。该表系统地概述了肠道微生物群组成的疾病特异性改变、关键机制途径和整个手稿中讨论的循证干预策略。
- 评论:表 1 过于简单化。它应该扩展以包含更多相关数据,或者如果它没有增加重要价值,则将其删除。
响应:
我们衷心感谢审阅者的深刻反馈。
根据您的建议,我们对表 1 (第 14 页,第 625 行) 进行了全面修订,即对之前建议的表 1 的回复,以提供更全面的总结本综述所涉及的代谢疾病。
- Comment:The topic of fecal microbiota transplantation is discussed only briefly. This subject deserves more in-depth analysis considering its therapeutic potential.
Response:
We sincerely thank the reviewer for this valuable suggestion.
In response, we have expanded the discussion on fecal microbiota transplantation (FMT) in Section 3.2.2.4 (Page 12, Lines 565) to comprehensively address its mechanisms, current clinical applications, and therapeutic potential in metabolic diseases. These additions enhance the scientific rigor of the discussion and provide readers with a more nuanced understanding of FMT's evolving role in metabolic disorder management.
- Comment:Greater emphasis should be placed on probiotics, particularly by discussing studies that highlight their limitations in fully restoring the gut microbiota.
Response:
We are grateful for this insightful suggestion.
Our revision now includes an analysis of the application of probiotics in Section 3.2.2.2 (Page 11, Line 490), analyzing human clinical and mouse model studies related to the treatment of metabolic diseases with probiotics in recent years. In addition, the limitations of current probiotic research were pointed out. These supplements strengthen the conceptual framework of the review by coordinating the mechanisms and implementation obstacles of microbiota therapy.
- Comment:In Chapter 4, it is recommended to address the importance of microRNAs in the context of metabolic syndrome. The following reference is useful: https://doi.org/10.3390/jcm14062054.
Response:We sincerely thank this valuable suggestion. To respond directly to your suggestions, we have integrated a specialized analysis of the role of microRNA in the pathogenesis of metabolic syndrome in Section 5 (Page 28, Lines 1135). This revision particularly focuses on their interaction with intestinal microbiota dysbiosis and their development potential.
- Comment:A dedicated Conclusions chapter should be added to succinctly summarize the main findings and their clinical relevance.
Response:
We appreciate this constructive suggestion.
作为直接回应,我们加强了第 5 节(第 29 页,第 1149 行)中的结论,以整合代谢紊乱、肠道微生物群动力学、益生菌干预和草药治疗之间的关键相互关系。修订文本明确了“微生物组 - 代谢 - 免疫”轴作为本文的框架,并整合了相关的临床治疗策略。
再次感谢您的建议。

Reviewer 2 Report
Comments and Suggestions for Authors
Dear Authors,
Thank you for the opportunity to review your manuscript, which addresses a timely and increasingly relevant topic: the interplay between gut microbiota, metabolic disorders, and probiotic-fermented herbal compounds. The overall approach is promising, and the manuscript offers valuable insights. However, I have identified several aspects that could be strengthened in terms of technical precision, conceptual depth, and structural coherence. The following comments are intended to support the improvement of your work and to assist its potential publication in a high-standard journal such as the International Journal of Molecular Sciences.
- Title
Consider specifying a more concrete temporal or therapeutic focus—for example, indicating whether the article is a systematic review, a therapeutic proposal, or a mechanistic analysis.
Suggested reformulation (optional):
Gut Microbiota-Targeted Therapeutics for Metabolic Disorders: Mechanistic Insights into the Synergy of Probiotic-Fermented Herbal Bioactives
- Abstract
- Lack of quantitative or specific data: although this is a review abstract, it would be beneficial to include examples of probiotic strains or bioactive compounds.
- Methodological ambiguity: it is not stated whether the review is systematic, narrative, or integrative.
- Limitations are too general: the abstract mentions “limitations and future directions” without specifying any.
- Potential thematic overreach: the breadth of topics addressed may create expectations that are difficult to fulfil in terms of depth.
The abstract would benefit from improved specificity, methodological clarity, and technical depth to meet the standards of a journal like IJMS. Including concrete examples of compounds and probiotics (which are indeed discussed in the body of the manuscript) would substantially enhance the proposal.
Introduction
While the manuscript provides relevant background information throughout the abstract and subsequent sections, it currently lacks a clearly defined “Introduction” section, which is expected in review articles submitted to the International Journal of Molecular Sciences.
The absence of a dedicated introduction weakens the structural integrity of the manuscript and limits the reader’s ability to understand the rationale, scope, and objectives of the review from the outset.
I strongly recommend the authors include an explicit Introduction section that:
- Clearly outlines the scientific and clinical relevance of gut microbiota-targeted therapy in metabolic disorders;
- Justifies the focus on probiotic-fermented herbal bioactives within this context;
- Defines the objectives and scope of the review;
- Briefly explains the methodological approach (e.g., narrative review, literature selection criteria);
- References recent and relevant literature to support the rationale.
This addition will enhance the clarity and academic rigour of the manuscript, aligning it with the journal’s standards for review articles.
1.1 Metabolic Disease Etiology and Status
- Clearly introduce the main metabolic disorders of interest: obesity, type 2 diabetes mellitus (T2DM), non-alcoholic fatty liver disease (NAFLD), hyperuricaemia, and osteoporosis.
- The term “etiology and status” may appear ambiguous or redundant; we suggest changing it to:
“Epidemiology and Clinical Significance of Metabolic Diseases”. - Add an introductory sentence that links this subsection with the gut–metabolism axis to maintain thematic coherence with the title.
- Include standardised global references such as those from the WHO or IDF to strengthen the validity of the data presented.
1.1.1 Obesity and T2DM
- Provides a global perspective, incorporating data from the World Obesity Atlas 2024 and the Global Burden of Disease.
- While prevalence is well described, the etiological links to the gut microbiota are not introduced, which is critical given the manuscript’s thematic focus.
- Include evidence linking dysbiosis to insulin resistance and visceral fat accumulation (e.g., Firmicutes/Bacteroidetes ratio, short-chain fatty acids [SCFAs], or lipopolysaccharides [LPS]).
- It would be preferable to summarise extensive statistics in a table or figure to facilitate readability.
1.1.2 Hyperlipidaemia
- The paragraph is overly general. It is recommended to introduce key biomarkers (LDL, HDL, triglycerides) and their association with gut microbiota alterations.
- Mention whether there is evidence of the impact of microbial metabolites such as secondary bile acids on lipid regulation.
- Briefly link to related hepatic disorders, such as NAFLD.
1.1.3 Hyperuricaemia and Gout
- References linking the gut microbiota to uric acid metabolism or excretion are missing (e.g., depletion of uricolytic bacteria).
- Suggest introducing authors who have explored the relationship between dysbiotic microbiota, systemic inflammation, and gout flare-ups.
General comments on Section 1.1 and its subdivisions
While these subsections are well-written from an epidemiological standpoint, they lack a strong connection to the manuscript’s central theme: the gut microbiota and its therapeutic implications. To strengthen coherence with the article’s focus, we recommend:
- Including pathophysiological mechanisms involving the microbiota from this early stage.
- Adding key references from clinical or experimental studies on dysbiosis and metabolism.
- Reformulating subsection headings, where necessary, to better reflect the relationship between microbiota and each disorder.
1.2. Treatment of Metabolic Diseases
- The role of AMPK and other relevant molecular mechanisms is appropriately mentioned, which aligns well with the molecular focus of the journal.
1.2.1. Lifestyle Interventions
- There is no explicit mention of microbiota interactions. It would be important to link these interventions to beneficial shifts in microbial composition and function, as discussed later in the manuscript.
- References are somewhat limited. Recent reviews on the impact of exercise and dietary patterns on the gut microbiota—and their relationship to insulin resistance and NAFLD—could strengthen the section.
- The gut–brain axis is omitted. Since mental health improvements are briefly noted, the inclusion of references regarding the microbiota's role in gut–brain modulation would be pertinent.
1.2.2. Drug Treatment
- Lacks connection to microbiota: Considering the manuscript's microbiota-centred approach, the discussion should address how certain drugs (e.g., metformin, GLP-1 analogues) modulate microbial composition and thereby improve metabolic outcomes.
- Imbalance between detail and relevance: While the discussion on Sortilin and ACSL1 is molecularly interesting, it appears somewhat peripheral to the manuscript's main focus on microbiota and fermented therapeutics. A brief mention would suffice, redirecting emphasis to microbiota-based strategies.
- Lack of clinical data or concrete examples: This section could be enhanced by citing clinical studies where pharmacological treatments have been associated with changes in microbial diversity or the abundance of key species such as Akkermansia muciniphila or Faecalibacterium prausnitzii.
- Specific writing suggestion: Consider including how widely-used metabolic drugs, such as metformin or GLP-1 analogues, not only improve glycaemic control but also alter gut microbial profiles, potentially amplifying therapeutic effects through microbiota–host interactions.
General recommendation for Section 1.2:
The section offers a broad and informative overview of current lifestyle and pharmacological strategies for metabolic disease management. However, given the manuscript’s core focus on microbiota modulation, I strongly recommend reinforcing the microbiota–host interaction perspective throughout both subsections. Linking therapeutic interventions to microbial modulation will enhance the manuscript's coherence and thematic integrity.
2.1. The Fundamental Role of Gut Microbiota in Disease and Health
- Lacks structural cohesion: This section merges general facts with specific conclusions without a clear logical progression. It would benefit from being reorganised into separate paragraphs that address:
- Genetic structure and microbial diversity;
- Physiological functions;
- Environmental and dietary factors affecting microbiota composition.
- Missing transition to the manuscript’s main topic: The therapeutic role of probiotics and fermented compounds is not introduced. The section should close with a paragraph justifying the relevance of microbiota modulation in the context of metabolic disorders.
2.1.1. Therapeutic Approaches for Diseases of the Digestive System
- The section effectively highlights links between microbiota and digestive diseases such as Crohn’s disease, colitis, and gastrointestinal side effects of cancer therapy.
- It accurately references microbiota and metabolite effects in cancer and mucositis therapy.
- However: The first example focuses on autism spectrum disorders (ASD), which are neurological rather than digestive conditions. Although the point is valid, it may confuse the subsection’s focus. It would be more appropriate to relocate this discussion to the neurological section.
- Greater depth is needed regarding specific therapeutic modulation, such as:
-Which probiotic strains have demonstrated efficacy?
-Which fermented compounds exert beneficial effects?
-What is the role of the gut barrier–microbiota axis?
2.1.2. Therapeutic Approaches for Diseases of the Neurological System
- This section remains overly general and does not address specific neurological conditions such as Alzheimer’s disease, depression, or Parkinson’s disease.
- It would be helpful to include clinical or preclinical studies showing tangible impacts of microbial modulation on neurological health, such as Lactobacillus rhamnosus in anxiety and stress models.
- The section lacks a concluding paragraph to summarise therapeutic implications and the translational challenges involved.
2.1.3. Therapeutic Approaches for Diseases of the Urinary System
- Requires more specificity and organisation: it discusses urinary tract infections, kidney transplantation, chronic kidney disease, and urological cancers without a clear hierarchy or thematic separation.
- It is recommended to divide the section into two thematic parts:
(1) Urinary tract infections;
(2) Chronic kidney disease and urological cancers. - The section would benefit from concrete therapeutic proposals, such as the use of targeted probiotic strains to reduce infection recurrence or to mitigate renal damage via microbial toxin modulation.
General Recommendation for Section 2.1 and Subsections:
The section provides valuable and current insights, but suffers from:
- Lack of a clearly defined thematic focus within subsections;
- Absence of concrete therapeutic examples (e.g., strains, compounds, outcomes);
- Need for a summary that integrates and transitions to subsequent content.
2.2.1.1. Obesity and T2DM
- The narrative conflates obesity and T2DM without clearly distinguishing their respective microbial profiles and pathophysiological mechanisms.
- Recent references concerning Faecalibacterium prausnitzii, Roseburia spp., and their protective roles are lacking.
- The contribution of increased intestinal permeability to insulin resistance should be more explicitly highlighted.
- Recommendation: Separate the discussion of obesity and T2DM into distinct subsections or clearly demarcated paragraphs. Incorporate recent clinical evidence and strengthen the discussion of immunometabolic mechanisms involved.
2.2.1.2. Hyperlipidaemia
- The subtitle should specify that the section primarily focuses on NAFLD as a model of dyslipidaemia.
- Bacterial species such as Klebsiella pneumoniae and Cholera spp. are mentioned without explanation of their specific roles or links to metabolic profiles.
- The involvement of gut microbiota in hepatic gene regulation (e.g., FXR, SREBP-1c) is not discussed.
- Recommendation: Clarify that hypertriglyceridaemia is addressed via the gut–liver axis, and strengthen the connection between microbial metabolites (e.g., ethanol, LPS) and hepatic lipogenesis.
2.2.1.3. Hyperuricaemia and Gout
- The section lacks deeper mechanistic insights into how microbiota alterations exacerbate hyperuricaemia (e.g., via intestinal permeability, systemic inflammation).
- It does not specify whether findings are consistent across human, animal or in vitro studies.
- There is no mention of whether probiotic-based therapies exist for the management of hyperuricaemia.
- Recommendation: Add a paragraph providing functional integration (e.g., microbiota–inflammation–renal protection axis), and relate it to current evidence on probiotic or prebiotic interventions.
2.2.2.1. Dietary Treatment
- This section is overly brief and general; no specific examples of dietary patterns (e.g., polyphenol-rich diets, Mediterranean or ketogenic diets) are provided despite their known impact on the microbiota.
- Detrimental dietary patterns (e.g., high-fat or high-sugar diets) are not critically addressed as a point of contrast.
- The section would benefit from more recent references (2023–2024), particularly clinical studies or meta-analyses on dietary interventions targeting the gut microbiota.
- Suggestion: Include specific examples of fermentable fibres and relate them to studies in human or animal models of metabolic disease.
2.2.2.2. Probiotics, Prebiotics and Postbiotics Treatment
- Although hyperuricaemia is mentioned, the clinical evidence for other metabolic disorders is not integrated in a systematic manner.
- The section blends physiological information with studies of varying evidence levels (in vitro, murine, human), without clear distinction.
- Potential limitations of postbiotics, including safety or bioavailability concerns, are not discussed.
- Suggestion: Consider adding a comparative table summarising the effects of selected probiotic strains versus known postbiotics. Include a critical reflection on the synergy between these compounds and dietary components.
2.2.2.3. Drug Treatment
- While microbiota-modulating effects of drugs are mentioned, a direct cause–effect relationship between microbiota changes and clinical outcomes is not consistently supported.
- The distinction between direct pharmacological effects and microbiota-mediated mechanisms is unclear.
- Recent literature examining longitudinal effects of drug treatments on the human microbiome (e.g., cohort studies or RCTs) should be incorporated.
- Suggestion: Expand this section to include data on interindividual variability in treatment response based on baseline microbiota composition, a key element in personalised medicine.
General Recommendation for Section 2.2.2:
This section provides a broad overview of strategies to modulate the gut microbiota in metabolic disorders. However, to meet the standards of IJMS, it is recommended to:
- Deepen the discussion of each intervention (diet, probiotics, pharmaceuticals) using specific and up-to-date evidence;
- Include distinct molecular mechanisms for each intervention type;
- Offer a critical discussion of limitations, challenges, and clinical perspectives;
- Consider using summary tables or mechanistic diagrams to enhance clarity and reader engagement.
3.1 – TCM for metabolic diseases
- The introductory paragraph of this section is overly general and lacks up-to-date key references regarding microbiota–TCM interactions in humans.
- A clearer delineation of the principal mechanism of action to be discussed in each subsection would be appreciated (e.g., effects on SCFAs, intestinal barrier, inflammatory pathways).
- It is recommended to incorporate clinical examples or early human trials, as the current focus is largely preclinical.
3.1.1 – TCM for obesity and T2DM
- Mechanistic depth is lacking: the section does not detail how these compounds modulate the gut microbiota or which bacterial taxa are involved.
- It is advised to add references to recent studies documenting the direct interaction between TCM compounds and gut microbiota in obesity/T2DM models (e.g., the role of Akkermansia muciniphila with berberine).
- Statements such as “TCM can treat obesity in many ways...” are overly general and lack mechanistic support or specific citations.
3.1.2 – TCM for NAFLD
- More detail is needed regarding the effects of these compounds on gut microbiota. For instance, how do these plants influence bacterial ethanol production, SCFA profiles, or hepatic gene expression?
- The clinical relevance of the compounds is unclear: is there evidence of human use or are findings limited to murine models?
- This subsection could benefit from an additional table or infographic summarising mechanisms and active ingredients of the TCM agents discussed.
3.1.3 – TCM for hyperuricaemia and gout
- This part could be strengthened by including microbial differences in gout patients, such as shifts in butyrate-producing bacteria or pro-inflammatory flora.
- Indicate whether there are clinical trials or current limitations to the real-world therapeutic application of TCM in this context.
- Include how fermented TCM products could enhance the bioavailability of compounds such as diosgenin.
Overall, section 3.1 and its subsections provide an insightful overview of the potential of Traditional Chinese Medicine in managing metabolic diseases, with a focus on microbiota. However, to meet the standards of a journal such as IJMS, it is essential to:
- Integrate more specific references to functional microbiota modulation.
- Deepen the discussion on molecular and microbial mechanisms.
- Include human data where available, or explicitly acknowledge the lack thereof.
3.2 – Extraction of active ingredients of TCM
- Lack of molecular or functional focus: While this section outlines extraction techniques, it does not explicitly connect them to the therapeutic functionality of the extracted compounds. It would be valuable to include examples linking bioactive compound types (e.g., flavonoids, alkaloids, polysaccharides) with relevant metabolic or immunomodulatory actions on the microbiota.
- Absence of selection criteria: Although multiple methods are described, there is no clear rationale for choosing one over another depending on compound type or intended application, which may reduce the practical utility for researchers.
- Poor integration with the manuscript’s main theme: The section is insufficiently linked to the central focus of the manuscript, which is the synergy between fermented herbal bioactives and probiotics. It would be beneficial to explain how the chosen extraction method may influence the bioavailability and fermentability of compounds by probiotics.
- Superficial technical language: Certain terms, such as “semimimetic extraction”, are not well defined or contextualised, which could confuse non-specialist readers.
Overall, while section 3.2 meets a general informative objective, it requires stronger conceptual integration with the probiotics–fermentation–microbiota axis, which is the manuscript’s core. Emphasising how extraction techniques impact intestinal functionality would improve the manuscript’s scientific cohesion.
3.3 – TCM Fermentation Methods
- Lack of a clear conceptual framework: Although the section refers to the benefits of fermentation, it does not sufficiently develop the biotechnological and microbiological context of these transformations, nor does it explicitly connect them to specific therapeutic goals.
- Limited integration with human metabolism or gut microbiota: The discussion lacks direct links between compound transformations and their effects on gut microbiota or host metabolic pathways—essential considerations in a manuscript focused on metabolic therapy.
- Imprecise language in some statements: Phrases such as “may improve efficacy” or “can generate new effects” are too vague. It is recommended to use evidence-based language such as “has been shown to...” or “increases the concentration of...”.
3.3.1 Traditional Fermentation
- Specific examples of traditionally fermented products and their clinical applications in Traditional Chinese Medicine (TCM) should be included.
- Recent references comparing the microbial activity of solid-state versus liquid-state fermentation are lacking.
3.3.2 Probiotic Fermentation
- This section would benefit from improved contextualisation in terms of the probiotic species employed, fermentation conditions, and the nature of fermentation-derived products.
- Batch-to-batch consistency and microbial standardisation are not addressed, yet these are critical for therapeutic reproducibility.
3.3.2.1 Enhancing Active Ingredients
- The discussion should contrast the actual bioavailability of bioactive compounds post-fermentation, not merely their increased concentration.
3.3.2.2 Reducing the Toxicity
- It would be valuable to include molecular detoxification mechanisms (e.g., specific enzymatic actions), rather than only reporting the final effect.
3.3.2.3 Generating New Compounds
- Further details are needed regarding the chemical nature and pharmacodynamic properties of newly generated compounds, and how these may be applied to metabolic diseases.
Section 3.3 addresses a highly relevant and increasingly studied topic. However, to meet the standards of IJMS, it is recommended to enhance technical precision, explicitly link to gut microbiota and metabolism, and deepen the molecular mechanisms underpinning the observed effects.
3.3.3 Fermentation Strains
- The section is largely descriptive; a more comparative approach is encouraged, including strain selection criteria (e.g., pH tolerance, enzymatic profiles, synergy with phytochemicals).
- The strain-specific immunomodulatory roles of certain probiotics are not discussed but would be relevant to the proposed therapeutic scope.
- Integration with other sections should be improved (e.g., highlighting how specific strains relate to the molecular mechanisms discussed later in the manuscript).
3.4.1 TCM as a Natural Prebiotic
- Key references regarding dose-dependent effects and clinical outcomes are missing.
- The heterogeneity of the human microbiome, potential adverse effects, and therapeutic limits are not addressed.
- A clearer distinction between in vivo and in vitro effects is needed, specifying whether data derive from animal models, human trials, or bacterial cultures.
3.4.2 TCM and the Intestinal Mucosal Barrier
- The discussion of molecular mechanisms remains superficial; references to specific tight junction proteins such as ZO-1 or occludin (although mentioned later) should be incorporated here.
- Adding the relationship between barrier integrity and inflammatory markers would enrich this section.
- Clinical trial data demonstrating a correlation between mucosal repair and metabolic improvements are notably absent.
3.4.3 TCM and Gut Microbiota Regulation
- This is one of the strongest sections. Key bacterial taxa such as Akkermansia muciniphila, Faecalibacterium prausnitzii, Roseburia, and SCFA-producing flora like Butyrivibrio are appropriately cited. Specific effects on glucose–lipid metabolism, insulin resistance, and diet-induced obesity are well discussed.
- However, no functional hierarchy or synergistic interactions between species are established.
- The section would be strengthened by integrating concepts such as enterotypes and interindividual microbiota variation.
- Expanding on hepatic metabolic effects and the gut–liver axis would also be valuable.
- Challenges and Future Directions
- Lack of critical prioritisation: While many challenges are listed, they are not ranked or weighed in terms of impact or urgency. The reader is left uncertain about the most significant bottlenecks to clinical translation.
- Scarce quantitative evidence: Despite broad coverage, no concrete examples or key studies are cited to support some major claims (e.g., recurrence of dysbiosis post-FMT or scalability limitations of solid fermentation extracts).
- Limited molecular depth: Although “multi-omics” technologies are mentioned, no specific biomarkers or metabolic pathways are discussed that could guide microbiota-based precision medicine.
- Lack of concrete innovation proposals: While there is general mention of “interdisciplinary collaboration and technological innovation”, it would be beneficial to propose specific models or platforms that foster such collaboration (e.g., international consortia, metagenomic data-sharing networks, or clinical biobanks with microbiota–host phenotyping).
I hope that the comments provided will prove helpful in refining your manuscript, and I encourage you to continue developing this important line of research with clarity and scientific rigour.
Respectfully
Author Response
1.Comment:
Title
- Consider specifying a more concrete temporal or therapeutic focus—for example, indicating whether the article is a systematic review, a therapeutic proposal, or a mechanistic analysis.
- Suggested reformulation (optional):
Gut Microbiota-Targeted Therapeutics for Metabolic Disorders: Mechanistic Insights into the Synergy of Probiotic-Fermented Herbal Bioactives
Response:
We sincerely appreciate this constructive suggestion.
Following your recommendation, we have revised the manuscript's title to "Gut Microbiota-Targeted Therapeutics for Metabolic Disorders: Mechanistic Insights into the Synergy of Probiotic-Fermented Herbal Bioactives" to more precisely encapsulate the study's dual focus on microbial modulation strategies and the therapeutic synergism between probiotics and herbal compounds.
2.Comment:
Abstract
- Lack of quantitative or specific data: although this is a review abstract, it would be beneficial to include examples of probiotic strains or bioactive compounds.
- Methodological ambiguity: it is not stated whether the review is systematic, narrative, or integrative.
- Limitations are too general: the abstract mentions “limitations and future directions” without specifying any.
- Potential thematic overreach: the breadth of topics addressed may create expectations that are difficult to fulfil in terms of depth.
Response:
We sincerely thank the reviewers for their insightful opinions on our manuscript.
In response to the feedback, we have added an introduction to better integrate probiotic fermentation with TCM , focusing on the fundamental principles of treating metabolic disorders in the gut microbiota. The added introduction part is on the first page of the article, line 39, and is marked in red. We further clarified the mechanism synergy between the biological activity of herbs and probiotics, emphasizing the improvement of bioavailability and microbial metabolic regulation. Thank you for your constructive suggestions, which have strengthened the focus of the manuscript on combining traditional herbal medicine with microbiome targeted therapy.
3.Comment:
1.1 Metabolic Disease Etiology and Status
- Clearly introduce the main metabolic disorders of interest: obesity, type 2 diabetes mellitus (T2DM), non-alcoholic fatty liver disease (NAFLD), hyperuricaemia, and osteoporosis.
- The term “etiology and status” may appear ambiguous or redundant; we suggest changing it to:
“Epidemiology and Clinical Significance of Metabolic Diseases”. - Add an introductory sentence that links this subsection with the gut–metabolism axis to maintain thematic coherence with the title.
- Include standardised global references such as those from the WHO or IDF to strengthen the validity of the data presented.
Response:
Thank you for your valuable suggestions.
In response, we have revised the title of Section 2.1 to "Epidemiology and Clinical Significance of Metabolic Diseases" to better align with the content. Additionally, in Section 2.1 (Page 3, Line 103), we incorporated a discussion on the interplay between metabolic diseases and the gut-metabolism axis to enhance thematic coherence. To strengthen the epidemiological data, standardized global references were added (Page 3, Line 99), ensuring the reliability of prevalence statistics. These revisions aim to improve clarity, accuracy, and alignment with the manuscript’s focus on gut microbiota-targeted therapeutics.
4.Comment:
1.1.1 Obesity and T2DM
- Provides a global perspective, incorporating data from the World Obesity Atlas 2024 and the Global Burden of Disease.
- While prevalence is well described, the etiological links to the gut microbiota are not introduced, which is critical given the manuscript’s thematic focus.
- Include evidence linking dysbiosis to insulin resistance and visceral fat accumulation (e.g., Firmicutes/Bacteroidetes ratio, short-chain fatty acids [SCFAs], or lipopolysaccharides [LPS]).
- It would be preferable to summarise extensive statistics in a table or figure to facilitate readability.
Response:
Thank you for your sincere opinion.
In response to your opinion, first of all, to better describe obesity and metabolic diseases, we have divided this into two parts, namely 2.1.1 and 2.1.2.
In these two sections, we have cited data on the 2024 World Obesity Atlas and the global Burden of Disease to more accurately reflect the epidemic trends and prevalence rates of obesity and T2DM. They are respectively on Page 3, Line 99 and Page 4, Line 112 of the article.
Secondly, in response to your second opinion, we respectively link obesity and T2DM to the intestinal metabolic axis to reflect the key points of this article. They are respectively on Page 3, Line 114 of Section 2.1.1 and Page 4, Line 136 of Section 2.1.2 of the article.
In addition, regarding the third opinion, while describing obesity, T2DM and the intestinal metabolic axis, we have added relevant metabolites such as SCFA or evidence related to microbiota dysbiosis (F/B value), etc., making these two sections more in line with the core theme of the article "Intestinal Microbiota-Host Metabolism". They are respectively on Page 3, Line 116 and Page 4, Line 138 of the article.
Finally, we organize the relevant data in these two sections into data to improve the readability of this article.
5.Comment:
1.1.2 Hyperlipidaemia
- The paragraph is overly general. It is recommended to introduce key biomarkers (LDL, HDL, triglycerides) and their association with gut microbiota alterations.
- Mention whether there is evidence of the impact of microbial metabolites such as secondary bile acids on lipid regulation.
- Briefly link to related hepatic disorders, such as NAFLD.
Response:
Thank you very much for your suggestions.
In response to the first suggestion, we have carefully revised the content of this section. In Section 2.1.3 of the article, on page 4, line 141, key biomarkers such as HDL, LDL and other related indicators have been introduced, and the changes of related indicators when hyperlipidemia occurs have been expounded, adding more professional knowledge to this article.
Regarding the second opinion, after carefully reviewing the relevant literature, the relationship between microbial metabolites and lipid regulation was linked, and knowledge related to metabolites such as secondary bile acids was added. The relationship between the intestinal microbiota and the occurrence of hyperlipidemia was emphasized. The added content is on page 4 of the article, line 158.
In response to your third suggestion, we have included NAFLD, a major disease related to hyperlipidemia, and elaborated on the information related to its incidence rate and hyperlipidemia, etc. The above content has further enriched the article's content.
6.Comment:
1.1.3 Hyperuricaemia and Gout
- References linking the gut microbiota to uric acid metabolism or excretion are missing (e.g., depletion of uricolytic bacteria).
- Suggest introducing authors who have explored the relationship between dysbiotic microbiota, systemic inflammation, and gout flare-ups.
Response:
Thank you very much for your suggestions.
Firstly, we have modified the content of this section by adding the global burden of disease data in Section 2.1.4, Page 5, Line 168 of the article to reflect the authenticity and completeness of the data.
In response to your first suggestion, we have added the content related to the gut microbiota and uric acid metabolism. In Section 2.1.4, page 4, line 171 of the article, it is pointed out that the occurrence of hyperuricemia and gout is closely related to the gut microbiota metabolism, which is in line with the theme of the article.
In response to your second suggestion, at the end of 2.1.4, on page 5, line 173, we have added relevant clinical studies on microbiota dysbiosis and gout attacks, strengthening the theoretical framework of the "microbiota-uric acid metabolism axis".
7.Comment:
General comments on Section 1.1 and its subdivisions
While these subsections are well-written from an epidemiological standpoint, they lack a strong connection to the manuscript’s central theme: the gut microbiota and its therapeutic implications. To strengthen coherence with the article’s focus, we recommend:
- Including pathophysiological mechanisms involving the microbiota from this early stage.
- Adding key references from clinical or experimental studies on dysbiosis and metabolism.
- Reformulating subsection headings, where necessary, to better reflect the relationship between microbiota and each disorder.
Response:
Thank you for your proposal.
In Section 2.1 of the manuscript, we added the relationship between the gut microbiota and metabolic diseases related to the theme of the manuscript. And in order to better fit the content of the article, literature related to clinical experimental studies has been added. Besides, we will elaborate on this relationship in detail in the third section later. For some of the title issues, to avoid confusion, we have divided obesity and diabetes into two subsections to elaborate on the relevant contents respectively.
8.Comment:
1.2.1. Lifestyle Interventions
- There is no explicit mention of microbiota interactions. It would be important to link these interventions to beneficial shifts in microbial composition and function, as discussed later in the manuscript.
- References are somewhat limited. Recent reviews on the impact of exercise and dietary patterns on the gut microbiota—and their relationship to insulin resistance and NAFLD—could strengthen the section.
- The gut–brain axis is omitted. Since mental health improvements are briefly noted, the inclusion of references regarding the microbiota's role in gut–brain modulation would be pertinent.
Response:
Thank you for your proposal.
For the first suggestion, we have added pertinent knowledge about beneficial changes such as intervention measures for treating metabolic diseases, such as exercise, to regulate the gut microbiota and increase the diversity of the flora. This part is in Section 2.2.1, page 5, line 193 of the article.
In response to your second suggestion, we have enhanced the reference materials related to the impact of exercise and dietary patterns on the gut microbiota, and increased the relevance of this section to the main content of the article. The added content is on page 5, on line 184 of the article.
Regarding the third suggestion, since the previous manuscript briefly mentioned the relevant content of "mental health", but since the content of this section mainly elaborates on the intervention measures of lifestyle for metabolic diseases, to avoid deviating from the topic, this content has now been deleted. Additional content that is more germane to the theme has been added.
9.Comment:
1.2.2. Drug Treatment
- Lacks connection to microbiota: Considering the manuscript's microbiota-centred approach, the discussion should address how certain drugs (e.g., metformin, GLP-1 analogues) modulate microbial composition and thereby improve metabolic outcomes.
- Imbalance between detail and relevance: While the discussion on Sortilin and ACSL1 is molecularly interesting, it appears somewhat peripheral to the manuscript's main focus on microbiota and fermented therapeutics. A brief mention would suffice, redirecting emphasis to microbiota-based strategies.
- Lack of clinical data or concrete examples: This section could be enhanced by citing clinical studies where pharmacological treatments have been associated with changes in microbial diversity or the abundance of key species such as Akkermansia muciniphila or Faecalibacterium prausnitzii.
- Specific writing suggestion: Consider including how widely-used metabolic drugs, such as metformin or GLP-1 analogues, not only improve glycaemic control but also alter gut microbial profiles, potentially amplifying therapeutic effects through microbiota–host interactions.
Response:
Thank you for your proposal.
Regarding your first suggestion. We have added relevant clinical studies on the influence of GLP-1 receptor agonists on metabolic diseases through the gastrointestinal microbiota in Section 2.2.2, page 6,line 222 of the article. And it has added the change of the third essential bacterial species, Akkermansia muciniphila, that you proposed.
Regarding the second content, since the Sortilin and ACSL1 therapies mentioned in the previous manuscript do not match the content of this section, this content has now been expunged, and the focus has been placed on the regulation of the gut microbiota.
Regarding the fourth content, since this section primarily elaborates on the relevant drug treatments for metabolic diseases nowadays, a brief discussion on drug treatments related to the gut microbiota is provided in this section. It will be emphasized in the following 3.2.2.5.
10.Comment:
General recommendation for Section 1.2:
The section offers a broad and informative overview of current lifestyle and pharmacological strategies for metabolic disease management. However, given the manuscript’s core focus on microbiota modulation, I strongly recommend reinforcing the microbiota–host interaction perspective throughout both subsections. Linking therapeutic interventions to microbial modulation will enhance the manuscript's coherence and thematic integrity.
Response:Thank you for your proposal. We carefully examined the manuscript itself and added more relevant treatment methods related to the gut microbiota that were in accordance with the theme. We will elucidate more realistically in Section 3.2.2.4 of the following article.
11.Comment:
2.1.1. Therapeutic Approaches for Diseases of the Digestive System
- The section effectively highlights links between microbiota and digestive diseases such as Crohn’s disease, colitis, and gastrointestinal side effects of cancer therapy.
- It accurately references microbiota and metabolite effects in cancer and mucositis therapy.
- However: The first example focuses on autism spectrum disorders (ASD), which are neurological rather than digestive conditions. Although the point is valid, it may confuse the subsection’s focus. It would be more appropriate to relocate this discussion to the neurological section.
- Greater depth is needed regarding specific therapeutic modulation, such as:
-Which probiotic strains have demonstrated efficacy?
-Which fermented compounds exert beneficial effects?
-What is the role of the gut barrier–microbiota axis?
Response:Thank you for your suggestion. Regarding your first suggestion, we have revised the content of the manuscript. Firstly, to avoid straying from the topic, we removed the section related to "autism spectrum disorder" from the article and added more diseases related to the digestive system, as well as the relationship between the gut microbiota and digestive system diseases.
In addition, in response to your second suggestion, we have added probiotic strains that have been proven effective in treating digestive system diseases and the role of the intestinal barrier - microbial axis. The importance of the gut microbiota was further emphasized.
12.Comment:
2.1.2. Therapeutic Approaches for Diseases of the Neurological System
- This section remains overly general and does not address specific neurologicalconditions such as Alzheimer’s disease, depression, or Parkinson’s disease.
- It would be helpful to include clinical or preclinical studies showing tangible impacts of microbial modulation on neurological health, such as Lactobacillus rhamnosus in anxiety and stress models.
- The section lacks a concluding paragraph to summarise therapeutic implications and the translational challenges involved.
Response:We appreciate this comment very much.In response to your proposal, we have revised the content of the article by adding the regulatory efficacy and mechanism of Lactobacillus rhamnosus on models such as anxiety and stress. In line 278 on page 7 of Section 3.1.2, it provides relevant literature support for the treatment of mental system diseases from the perspective of gut microbiota. And on line 291 on page 7 of Section 3.1.2, the potential of treating neurological diseases through the gut microbiota at present has been added.
13.Comment:
2.1.3. Therapeutic Approaches for Diseases of the Urinary System
- Requires more specificity and organisation: it discusses urinary tract infections, kidney transplantation, chronic kidney disease, and urological cancers without a clear hierarchy or thematic separation.
- It is recommended to divide the section into two thematic parts:
(1) Urinary tract infections;
(2) Chronic kidney disease and urological cancers. - The section would benefit from concrete therapeutic proposals, such as the use of targeted probiotic strains to reduce infection recurrence or to mitigate renal damage via microbial toxin modulation.
Response:
Thank you for your suggestions. In order to accurately clarify the hierarchical structure of this section, we have divided the content of this section into two parts.
The first part elaborates on the level of urinary tract infection. It has increased the mechanism of action of probiotics in urinary tract infections and their potential health benefits. The additional content is on line 304 on page 7 of 3.1.3.1 of the article.
The second part elaborates on chronic kidney disease and urinary system cancers, and adds the potential role of probiotics in the treatment of chronic kidney disease and enhancing immunotherapy. The additional content is on line 323 on page 7 of Section 3.1.3.2 of the article. All the added content is displayed in red font.
14.Comment:
General recommendation for Section 1.2:
The section offers a broad and informative overview of current lifestyle and pharmacological strategies for metabolic disease management. However, given the manuscript’s core focus on microbiota modulation, I strongly recommend reinforcing the microbiota–host interaction perspective throughout both subsections. Linking therapeutic interventions to microbial modulation will enhance the manuscript's coherence and thematic integrity.
Response:
Thank you for your suggestion.
By resetting the subheadings, the content of this section has been strengthened, and application examples of probiotics and other microbiota in treating diseases have been added, providing a more solid foundation for the treatment of diseases by intestinal microbiota.
15.Comment:
2.2.1.1. Obesity and T2DM
- The narrative conflates obesity and T2DM without clearly distinguishing their respective microbial profiles and pathophysiological mechanisms.
- Recent references concerning Faecalibacterium prausnitzii, Roseburia spp., and their protective roles are lacking.
- The contribution of increased intestinal permeability to insulin resistance should be more explicitly highlighted.
- Recommendation: Separate the discussion of obesity and T2DM into distinct subsections or clearly demarcated paragraphs. Incorporate recent clinical evidence and strengthen the discussion of immunometabolic mechanisms involved.
Response:
Thank you for your detailed suggestions. We have optimized this section.
First of all, in order to avoid confusing obesity with T2DM, we have divided this section into two parts: 3.2.1.1 Obesity and 3.2.1.2 T2DM. Secondly, we discussed the main mechanism of obesity in line 361 on page 8 of Section 3.2.1.1 of the article.
In addition, we discussed the influence of intestinal permeability on insulin resistance in line 379 on page 9 of Section 3.2.2.2 of the article. These mechanisms reveal the crucial role of the gut microbiota in metabolic diseases. The modified content is marked in red in the article.
16.Comment:
2.2.1.2. Hyperlipidaemia
- The subtitle should specify that the section primarily focuses on NAFLD as a model of dyslipidaemia.
- Bacterial species such as Klebsiella pneumoniae and Cholera spp. are mentioned without explanation of their specific roles or links to metabolic profiles.
- The involvement of gut microbiota in hepatic gene regulation (e.g., FXR, SREBP-1c) is not discussed.
- Recommendation: Clarify that hypertriglyceridaemia is addressed via the gut–liver axis, and strengthen the connection between microbial metabolites (e.g., ethanol, LPS) and hepatic lipogenesis.
Response:
Thank you for your professional guidance.
Regarding your first suggestion. The content of this section is divided into two sections for separate elaboration, namely 3.2.1.3 Hyperlipidemia and 3.2.1.4 NAFLD.
Regarding your second suggestion, the influence of Klebsiella pneumoniae and cholera bacteria on the development of NAFLD is expounded in line 413 on page 9 and line 424 on page 19 of Section 3.2.1.4.
Regarding your third suggestion, the genes related to the regulation of liver metabolism by the gut microbiota were profoundly discussed in line 403 on page 9 of Section 3.2.1.3 of the article. It reveals the role of intestinal bacteria at the molecular level.
Regarding your fourth suggestion, it is elaborated in 3.2.1.3, strengthening the connection between microbial metabolites and liver adipogenesis.
17.Comment:
2.2.1.3. Hyperuricaemia and Gout
- The section lacks deeper mechanistic insights into how microbiota alterations exacerbate hyperuricaemia (e.g., via intestinal permeability, systemic inflammation).
- It does not specify whether findings are consistent across human, animal or in vitro studies.
- There is no mention of whether probiotic-based therapies exist for the management of hyperuricaemia.
- Recommendation: Add a paragraph providing functional integration (e.g., microbiota–inflammation–renal protection axis), and relate it to current evidence on probiotic or prebiotic interventions.
Response:
Thank you for your inspiration.
In line 449 on page 10 of Section 3.2.1.5 of this article, the LPS produced by Gram-negative bacteria is described, which leads to a series of chain reactions in the body. It has been proved that microorganisms can aggravate hyperuricemia through their metabolites.
Regarding the relevant probiotic therapy, we will elaborate in detail on line 490 on page 11 of Section 3.2.2.2.
18.Comment:
2.2.2.1. Dietary Treatment
- This section is overly brief and general; no specific examples of dietary patterns (e.g., polyphenol-rich diets, Mediterranean or ketogenic diets) are provided despite their known impact on the microbiota.
- Detrimental dietary patterns (e.g., high-fat or high-sugar diets) are not critically addressed as a point of contrast.
- The section would benefit from more recent references (2023–2024), particularly clinical studies or meta-analyses on dietary interventions targeting the gut microbiota.
- Suggestion: Include specific examples of fermentable fibres and relate them to studies in human or animal models of metabolic disease.
Response:
Thank you for your proposal.
Regarding your first proposal, we elaborated on the impact of the Mediterranean diet and the ketogenic diet on the microbiome on page 11, line 474 of Section 3.2.2.1 of the article. These examples illustrate that different types of diets have different effects on the function of the gut microbiota.
Regarding your second suggestion. We elaborate on the dietary patterns of obese patients on line 485 on page 11. Additionally, the third suggestion of fermentable fibers has been added, revealing the need for deeper exploration and research in the field of fermentable fibers in the future.
19.Comment:
2.2.2.2. Probiotics, Prebiotics and Postbiotics Treatment
- Although hyperuricaemia is mentioned, the clinical evidence for other metabolic disorders is not integrated in a systematic manner.
- The section blends physiological information with studies of varying evidence levels (in vitro, murine, human), without clear distinction.
- Potential limitations of postbiotics, including safety or bioavailability concerns, are not discussed.
- Suggestion: Consider adding a comparative table summarising the effects of selected probiotic strains versus known postbiotics. Include a critical reflection on the synergy between these compounds and dietary components.
Response:
Thank you for the proposal.
Regarding your first suggestion. We elaborated on line 499 on page 11 of Section 3.2.2.2 of the article, connecting with other clinical evidence related to metabolic diseases and using multiple clinical studies to prove the availability and effectiveness of probiotics.
Regarding the second suggestion, we separate the clinical research and the mouse research within the paragraph to avoid confusion.
Regarding your third proposal, we separate probiotics from prebiotics and postbiotics as 3.2.2.2 and 3.2.2.3. In line 558 on page 12 of Section 3.2.2.3, the potential limitations of postbiotics were carefully explored, indicating that the research on postbiotics needs to be more in-depth to address their potential risks and optimize their application value.
In response to your fourth proposal, we have added Table 1 (page 14), whose content includes probiotics, postbiotics, and comparisons between human experiments and mouse experiments.
20.Comment:
2.2.2.3. Drug Treatment
- While microbiota-modulating effects of drugs are mentioned, a direct cause–effect relationship between microbiota changes and clinical outcomes is not consistently supported.
- The distinction between direct pharmacological effects and microbiota-mediated mechanisms is unclear.
- Recent literature examining longitudinal effects of drug treatments on the human microbiome (e.g., cohort studies or RCTs) should be incorporated.
- Suggestion: Expand this section to include data on interindividual variability in treatment response based on baseline microbiota composition, a key element in personalised medicine.
Response:
Thank you for your professional guidance.
By reading the literature, we described the direct effects of the drugs, such as liraglutide, etc., which have no obvious effect on the gut microbiota, in line 615 on page 13 of Section 3.2.2.5 of the article. Meanwhile, examples of the baseline characteristics of the intestinal microbiota affecting the therapeutic effect were added. On line 602 on page 13 of the article, it was proved that it is a key factor in individualized medicine.
21.Comment:
3.1 – TCM for metabolic diseases
- The introductory paragraph of this section is overly general and lacks up-to-date key references regarding microbiota–TCM interactions in humans.
- A clearer delineation of the principal mechanism of action to be discussed in each subsection would be appreciated (e.g., effects on SCFAs, intestinal barrier, inflammatory pathways).
- It is recommended to incorporate clinical examples or early human trials, as the current focus is largely preclinical.
Response:
Thank you for the proposal.
In response to your proposal, I have incorporated the content about the interaction between the human microbiota and TCM into different paragraphs, providing detailed descriptions of each disease within the paragraphs. The mechanism of action of its TCM is described in detail in each subsection.
22.Comment:
3.1.1 – TCM for obesity and T2DM
- Mechanistic depth is lacking: the section does not detail how these compounds modulate the gut microbiota or which bacterial taxa are involved.
- It is advised to add references to recent studies documenting the direct interaction between TCM compounds and gut microbiota in obesity/T2DM models (e.g., the role of Akkermansia muciniphila with berberine).
- Statements such as “TCM can treat obesity in many ways...” are overly general and lack mechanistic support or specific citations.
Response:
Thank you for your detailed proposal.
In this section, in order to avoid confusing Obesity with T2DM, this is divided into two parts, namely 4.1.1 T2DM and 4.1.2 Obesity.
Regarding your first suggestion, in line 641 on page 18 of Section 4.1.1, a detailed elaboration is provided on the regulation of gut microbiota by different TCM compounds for the treatment of T2DM.
Regarding the second suggestion, the role of TCM compounds and gut microbiota in obese patients is elaborated in detail in line 675 on page 18 of Section 4.1.2.
Regarding the third suggestion, some sentences in the article have been modified.
23.Comment:
3.1.2 – TCM for NAFLD
- More detail is needed regarding the effects of these compounds on gut microbiota. For instance, how do these plants influence bacterial ethanol production, SCFA profiles, or hepatic gene expression?
- The clinical relevance of the compounds is unclear: is there evidence of human use or are findings limited to murine models?
- This subsection could benefit from an additional table or infographic summarising mechanisms and active ingredients of the TCM agents discussed.
Response:
Thank you for your guidance.
Firstly, since NAFLD is one of the dyslipidemia models, NAFLD is regarded as the main focus of this section, and the title of the article is changed to "Hyperlipidemia".
In response to your suggestions, in this section, we have added that TCM molecules influence the gut microbiota through different mechanisms (such as the genetic level and the molecular level) to treat hyperlipidemia and NAFLD, etc. The additional content is on line 703 on page 19 of Section 4.1.3.
In addition, this section can benefit from Table 2 (page 20, line 763), which summarizes the TCM and active ingredients discussed.
24.Comment:
3.1.3 – TCM for hyperuricaemia and gout
- This part could be strengthened by including microbial differences in gout patients, such as shifts in butyrate-producing bacteria or pro-inflammatory flora.
- Indicate whether there are clinical trials or current limitations to the real-world therapeutic application of TCM in this context.
- Include how fermented TCM products could enhance the bioavailability of compounds such as diosgenin.
Response:
Thank you for your guidance.
In this section, we have strengthened the research on the limitations of the therapeutic effects of TCM on hyperuricemia and gout.The added content is on line 751 on page 20
25.Comment:
3.2 – Extraction of active ingredients of TCM
- Lack of molecular or functional focus: While this section outlines extraction techniques, it does not explicitly connect them to the therapeutic functionality of the extracted compounds. It would be valuable to include examples linking bioactive compound types (e.g., flavonoids, alkaloids, polysaccharides) with relevant metabolic or immunomodulatory actions on the microbiota.
- Absence of selection criteria: Although multiple methods are described, there is no clear rationale for choosing one over another depending on compound type or intended application, which may reduce the practical utility for researchers.
- Poor integration with the manuscript’s main theme: The section is insufficiently linked to the central focus of the manuscript, which is the synergy between fermented herbal bioactives and probiotics. It would be beneficial to explain how the chosen extraction method may influence the bioavailability and fermentability of compounds by probiotics.
- Superficial technical language: Certain terms, such as “semimimetic extraction”, are not well defined or contextualised, which could confuse non-specialist readers.
Response:
Thank you for your sincere suggestion.
In response to your first suggestion, we have revised the content of the manuscript. In line 793 on page 21 of Section 4.2 of the article, relevant contents such as how the molecules of TCM compounds are transformed after fermentation and how their therapeutic effects are improved (such as flavonoids, etc.) have been added.
In response to your second suggestion, we have added relevant knowledge in the article (21 pages, 794 lines) that different extraction methods have different effects, expanding the professional knowledge related to the manuscript.
In response to your third suggestion, we have modified and deleted some superficial language.
26.Comment:
3.3 – TCM Fermentation Methods
- Lack of a clear conceptual framework: Although the section refers to the benefits of fermentation, it does not sufficiently develop the biotechnological and microbiological context of these transformations, nor does it explicitly connect them to specific therapeutic goals.
- Limited integration with human metabolism or gut microbiota: The discussion lacks direct links between compound transformations and their effects on gut microbiota or host metabolic pathways—essential considerations in a manuscript focused on metabolic therapy.
- Imprecise language in some statements: Phrases such as “may improve efficacy” or “can generate new effects” are too vague. It is recommended to use evidence-based language such as “has been shown to...” or “increases the concentration of...”.
3.3.1 Traditional Fermentation
- Specific examples of traditionally fermented products and their clinical applications in TCMshould be included.
- Recent references comparing the microbial activity of solid-state versus liquid-state fermentation are lacking.
Response:
Thank you very much for your careful guidance to us.
In response to your suggestions, we have made modifications to Part 4.3 and 4.3.1. The inaccurate statements in the text have been deleted and modified.
In the modifications of the following sections, pay attention to the connection between the compounds produced and the host metabolism.
In 4.3.1, for your first suggestion, specific examples of traditional fermentation products and their applications have been added on line 814 on page 22.
Regarding your second suggestion, on line 825 on page 22, a comparison of the microbial activities in solid-state and liquid fermentation was made.
27.Comment:
3.3.2 Probiotic Fermentation
- This section would benefit from improved contextualisation in terms of the probiotic species employed, fermentation conditions, and the nature of fermentation-derived products.
- Batch-to-batch consistency and microbial standardisation are not addressed, yet these are critical for therapeutic reproducibility.
Response:
Thank you for your guidance.
In line 847 on page 22 of Section 4.3.2 of the text, we have added the elaboration on the production, screening and standardization of probiotics. Such as requiring the screening of the characteristics of the strain itself and ensuring the final safety assessment, etc.
28.Comment:
3.3.2.1 Enhancing Active Ingredients
The discussion should contrast the actual bioavailability of bioactive compounds post-fermentation, not merely their increased concentration.
Response:
Thank you for your guidance.
Regarding your suggestion. In line 874 on page 23 and line 883 on page 23 of Section 4.3.2.1 of the article, examples of the increased practical utilization of active ingredients after probiotic fermentation have been added.
29.Comment:
3.3.2.2 Reducing the Toxicity
It would be valuable to include molecular detoxification mechanisms (e.g., specific enzymatic actions), rather than only reporting the final effect.
Response:
Thank you for your suggestion.
In response to your suggestion, the relevant molecular mechanisms and examples of detoxification of certain components in TCM after probiotic fermentation have been added on line 891 on page 23 of Section 4.3.2.2 of the article.
30.Comment:
3.3.2.3 Generating New Compounds
Further details are needed regarding the chemical nature and pharmacodynamic properties of newly generated compounds, and how these may be applied to metabolic diseases.
Response:
Thank you for your suggestion.
In response to your suggestion, the relevant molecular mechanisms and examples of detoxification of certain components in TCM after probiotic fermentation have been added on line 891 on page 23 of Section 4.3.2.2 of the article.
31.Comment:
3.3.3 Fermentation Strains
- The section is largely descriptive; a more comparative approach is encouraged, including strain selection criteria (e.g., pH tolerance, enzymatic profiles, synergy with phytochemicals).
- The strain-specific immunomodulatory roles of certain probiotics are not discussed but would be relevant to the proposed therapeutic scope.
- Integration with other sections should be improved (e.g., highlighting how specific strains relate to the molecular mechanisms discussed later in the manuscript).
Response:
Thank you for your careful guidance.
In response to your suggestion, we have added the selection criteria for probiotics when choosing fermentation products in line 926 on page 24 of Section 4.3.3 of the article. Including PH tolerance, etc.
In line 925 on page 25, it has been added that by comparing the different characteristics of the strains and their interactions with the components of TCM, the therapeutic effect can be better exerted.
32.Comment:
3.4.1 TCM as a Natural Prebiotic
- Key references regarding dose-dependent effects and clinical outcomes are missing.
- The heterogeneity of the human microbiome, potential adverse effects, and therapeutic limits are not addressed.
- A clearer distinction between in vivo and in vitro effects is needed, specifying whether data derive from animal models, human trials, or bacterial cultures.
Response:
Thank you for your suggestion.
Regarding your first suggestion, in line 977 on page 25 of Section 4.4.1 of the article, the literature support related to the regulatory effect of TCM on the gut microbiota and the dose-effect relationship of clinical efficacy within a certain dose range has been added, illustrating the application of dose-dependent medicine in TCM.
In response to your second suggestion, we have added knowledge content about the relevant limiting factors such as the heterogeneity of the human microbiome and the expected individual differences in related clinical outcomes in line 985 on page 25 of the article.
Regarding your third suggestion. We explored the limitations of current experiments focusing on human experiments. When adopting traditional Chinese medical therapies, multiple factors need to be considered. (25 pages, 989 lines)
33.Comment:
3.4.2 TCM and the Intestinal Mucosal Barrier
- The discussion of molecular mechanisms remains superficial; references to specific tight junction proteins such as ZO-1 or occludin (although mentioned later) should be incorporated here.
- Adding the relationship between barrier integrity and inflammatory markers would enrich this section.
- Clinical trial data demonstrating a correlation between mucosal repair and metabolic improvements are notably absent.
Response:
Thank you for your careful guidance.
In response to your first suggestion, we have added relevant knowledge points about tight junction proteins on page 26, line 1019 of the article.
In response to your second suggestion, we have added literature support and content related to inflammatory markers and the integrity of intestinal barrier function on page 26 and 999 lines.
Regarding your third suggestion, we have elaborated in the paragraph on the improvement of metabolism after mucosal repair and the related content of improving metabolic diseases.
34.Comment:
4.Challenges and Future Directions
- Lack of critical prioritisation: While many challenges are listed, they are not ranked or weighed in terms of impact or urgency. The reader is left uncertain about the most significant bottlenecks to clinical translation.
- Scarce quantitative evidence: Despite broad coverage, no concrete examples or key studies are cited to support some major claims (e.g., recurrence of dysbiosis post-FMT or scalability limitations of solid fermentation extracts).
- Limited molecular depth: Although “multi-omics” technologies are mentioned, no specific biomarkers or metabolic pathways are discussed that could guide microbiota-based precision medicine.
- Lack of concrete innovation proposals: While there is general mention of “interdisciplinary collaboration and technological innovation”, it would be beneficial to propose specific models or platforms that foster such collaboration (e.g., international consortia, metagenomic data-sharing networks, or clinical biobanks with microbiota–host phenotyping).
Response:
Thank you for your suggestion.
In Section 5, page 28, page 1130 of the article, we have added relevant knowledge such as the research on multi-omics technology - metagenomic technology, and conducted relevant discussions on micro RNA.
Thank you again for the suggestion.
Dear reviewer:
I would like to express my deep gratitude to you for the constructive comments and suggestions.
A Point-by-Point Response to reviewers’ comments is given below.And all modifications are highlighted in red.
1.Comment:
Title
- Consider specifying a more concrete temporal or therapeutic focus—for example, indicating whether the article is a systematic review, a therapeutic proposal, or a mechanistic analysis.
- Suggested reformulation (optional):
Gut Microbiota-Targeted Therapeutics for Metabolic Disorders: Mechanistic Insights into the Synergy of Probiotic-Fermented Herbal Bioactives
Response:
We sincerely appreciate this constructive suggestion.
Following your recommendation, we have revised the manuscript's title to "Gut Microbiota-Targeted Therapeutics for Metabolic Disorders: Mechanistic Insights into the Synergy of Probiotic-Fermented Herbal Bioactives" to more precisely encapsulate the study's dual focus on microbial modulation strategies and the therapeutic synergism between probiotics and herbal compounds.
2.Comment:
Abstract
- Lack of quantitative or specific data: although this is a review abstract, it would be beneficial to include examples of probiotic strains or bioactive compounds.
- Methodological ambiguity: it is not stated whether the review is systematic, narrative, or integrative.
- Limitations are too general: the abstract mentions “limitations and future directions” without specifying any.
- Potential thematic overreach: the breadth of topics addressed may create expectations that are difficult to fulfil in terms of depth.
Response:
We sincerely thank the reviewers for their insightful opinions on our manuscript.
In response to the feedback, we have added an introduction to better integrate probiotic fermentation with TCM , focusing on the fundamental principles of treating metabolic disorders in the gut microbiota. The added introduction part is on the first page of the article, line 39, and is marked in red. We further clarified the mechanism synergy between the biological activity of herbs and probiotics, emphasizing the improvement of bioavailability and microbial metabolic regulation. Thank you for your constructive suggestions, which have strengthened the focus of the manuscript on combining traditional herbal medicine with microbiome targeted therapy.
3.Comment:
1.1 Metabolic Disease Etiology and Status
- Clearly introduce the main metabolic disorders of interest: obesity, type 2 diabetes mellitus (T2DM), non-alcoholic fatty liver disease (NAFLD), hyperuricaemia, and osteoporosis.
- The term “etiology and status” may appear ambiguous or redundant; we suggest changing it to:
“Epidemiology and Clinical Significance of Metabolic Diseases”. - Add an introductory sentence that links this subsection with the gut–metabolism axis to maintain thematic coherence with the title.
- Include standardised global references such as those from the WHO or IDF to strengthen the validity of the data presented.
Response:
Thank you for your valuable suggestions.
In response, we have revised the title of Section 2.1 to "Epidemiology and Clinical Significance of Metabolic Diseases" to better align with the content. Additionally, in Section 2.1 (Page 3, Line 103), we incorporated a discussion on the interplay between metabolic diseases and the gut-metabolism axis to enhance thematic coherence. To strengthen the epidemiological data, standardized global references were added (Page 3, Line 99), ensuring the reliability of prevalence statistics. These revisions aim to improve clarity, accuracy, and alignment with the manuscript’s focus on gut microbiota-targeted therapeutics.
4.Comment:
1.1.1 Obesity and T2DM
- Provides a global perspective, incorporating data from the World Obesity Atlas 2024 and the Global Burden of Disease.
- While prevalence is well described, the etiological links to the gut microbiota are not introduced, which is critical given the manuscript’s thematic focus.
- Include evidence linking dysbiosis to insulin resistance and visceral fat accumulation (e.g., Firmicutes/Bacteroidetes ratio, short-chain fatty acids [SCFAs], or lipopolysaccharides [LPS]).
- It would be preferable to summarise extensive statistics in a table or figure to facilitate readability.
Response:
Thank you for your sincere opinion.
In response to your opinion, first of all, to better describe obesity and metabolic diseases, we have divided this into two parts, namely 2.1.1 and 2.1.2.
In these two sections, we have cited data on the 2024 World Obesity Atlas and the global Burden of Disease to more accurately reflect the epidemic trends and prevalence rates of obesity and T2DM. They are respectively on Page 3, Line 99 and Page 4, Line 112 of the article.
Secondly, in response to your second opinion, we respectively link obesity and T2DM to the intestinal metabolic axis to reflect the key points of this article. They are respectively on Page 3, Line 114 of Section 2.1.1 and Page 4, Line 136 of Section 2.1.2 of the article.
In addition, regarding the third opinion, while describing obesity, T2DM and the intestinal metabolic axis, we have added relevant metabolites such as SCFA or evidence related to microbiota dysbiosis (F/B value), etc., making these two sections more in line with the core theme of the article "Intestinal Microbiota-Host Metabolism". They are respectively on Page 3, Line 116 and Page 4, Line 138 of the article.
Finally, we organize the relevant data in these two sections into data to improve the readability of this article.
5.Comment:
1.1.2 Hyperlipidaemia
- The paragraph is overly general. It is recommended to introduce key biomarkers (LDL, HDL, triglycerides) and their association with gut microbiota alterations.
- Mention whether there is evidence of the impact of microbial metabolites such as secondary bile acids on lipid regulation.
- Briefly link to related hepatic disorders, such as NAFLD.
Response:
Thank you very much for your suggestions.
In response to the first suggestion, we have carefully revised the content of this section. In Section 2.1.3 of the article, on page 4, line 141, key biomarkers such as HDL, LDL and other related indicators have been introduced, and the changes of related indicators when hyperlipidemia occurs have been expounded, adding more professional knowledge to this article.
Regarding the second opinion, after carefully reviewing the relevant literature, the relationship between microbial metabolites and lipid regulation was linked, and knowledge related to metabolites such as secondary bile acids was added. The relationship between the intestinal microbiota and the occurrence of hyperlipidemia was emphasized. The added content is on page 4 of the article, line 158.
In response to your third suggestion, we have included NAFLD, a major disease related to hyperlipidemia, and elaborated on the information related to its incidence rate and hyperlipidemia, etc. The above content has further enriched the article's content.
6.Comment:
1.1.3 Hyperuricaemia and Gout
- References linking the gut microbiota to uric acid metabolism or excretion are missing (e.g., depletion of uricolytic bacteria).
- Suggest introducing authors who have explored the relationship between dysbiotic microbiota, systemic inflammation, and gout flare-ups.
Response:
Thank you very much for your suggestions.
Firstly, we have modified the content of this section by adding the global burden of disease data in Section 2.1.4, Page 5, Line 168 of the article to reflect the authenticity and completeness of the data.
In response to your first suggestion, we have added the content related to the gut microbiota and uric acid metabolism. In Section 2.1.4, page 4, line 171 of the article, it is pointed out that the occurrence of hyperuricemia and gout is closely related to the gut microbiota metabolism, which is in line with the theme of the article.
In response to your second suggestion, at the end of 2.1.4, on page 5, line 173, we have added relevant clinical studies on microbiota dysbiosis and gout attacks, strengthening the theoretical framework of the "microbiota-uric acid metabolism axis".
7.Comment:
General comments on Section 1.1 and its subdivisions
While these subsections are well-written from an epidemiological standpoint, they lack a strong connection to the manuscript’s central theme: the gut microbiota and its therapeutic implications. To strengthen coherence with the article’s focus, we recommend:
- Including pathophysiological mechanisms involving the microbiota from this early stage.
- Adding key references from clinical or experimental studies on dysbiosis and metabolism.
- Reformulating subsection headings, where necessary, to better reflect the relationship between microbiota and each disorder.
Response:
Thank you for your proposal.
In Section 2.1 of the manuscript, we added the relationship between the gut microbiota and metabolic diseases related to the theme of the manuscript. And in order to better fit the content of the article, literature related to clinical experimental studies has been added. Besides, we will elaborate on this relationship in detail in the third section later. For some of the title issues, to avoid confusion, we have divided obesity and diabetes into two subsections to elaborate on the relevant contents respectively.
8.Comment:
1.2.1. Lifestyle Interventions
- There is no explicit mention of microbiota interactions. It would be important to link these interventions to beneficial shifts in microbial composition and function, as discussed later in the manuscript.
- References are somewhat limited. Recent reviews on the impact of exercise and dietary patterns on the gut microbiota—and their relationship to insulin resistance and NAFLD—could strengthen the section.
- The gut–brain axis is omitted. Since mental health improvements are briefly noted, the inclusion of references regarding the microbiota's role in gut–brain modulation would be pertinent.
Response:
Thank you for your proposal.
For the first suggestion, we have added pertinent knowledge about beneficial changes such as intervention measures for treating metabolic diseases, such as exercise, to regulate the gut microbiota and increase the diversity of the flora. This part is in Section 2.2.1, page 5, line 193 of the article.
In response to your second suggestion, we have enhanced the reference materials related to the impact of exercise and dietary patterns on the gut microbiota, and increased the relevance of this section to the main content of the article. The added content is on page 5, on line 184 of the article.
Regarding the third suggestion, since the previous manuscript briefly mentioned the relevant content of "mental health", but since the content of this section mainly elaborates on the intervention measures of lifestyle for metabolic diseases, to avoid deviating from the topic, this content has now been deleted. Additional content that is more germane to the theme has been added.
9.Comment:
1.2.2. Drug Treatment
- Lacks connection to microbiota: Considering the manuscript's microbiota-centred approach, the discussion should address how certain drugs (e.g., metformin, GLP-1 analogues) modulate microbial composition and thereby improve metabolic outcomes.
- Imbalance between detail and relevance: While the discussion on Sortilin and ACSL1 is molecularly interesting, it appears somewhat peripheral to the manuscript's main focus on microbiota and fermented therapeutics. A brief mention would suffice, redirecting emphasis to microbiota-based strategies.
- Lack of clinical data or concrete examples: This section could be enhanced by citing clinical studies where pharmacological treatments have been associated with changes in microbial diversity or the abundance of key species such as Akkermansia muciniphila or Faecalibacterium prausnitzii.
- Specific writing suggestion: Consider including how widely-used metabolic drugs, such as metformin or GLP-1 analogues, not only improve glycaemic control but also alter gut microbial profiles, potentially amplifying therapeutic effects through microbiota–host interactions.
Response:
Thank you for your proposal.
Regarding your first suggestion. We have added relevant clinical studies on the influence of GLP-1 receptor agonists on metabolic diseases through the gastrointestinal microbiota in Section 2.2.2, page 6,line 222 of the article. And it has added the change of the third essential bacterial species, Akkermansia muciniphila, that you proposed.
Regarding the second content, since the Sortilin and ACSL1 therapies mentioned in the previous manuscript do not match the content of this section, this content has now been expunged, and the focus has been placed on the regulation of the gut microbiota.
Regarding the fourth content, since this section primarily elaborates on the relevant drug treatments for metabolic diseases nowadays, a brief discussion on drug treatments related to the gut microbiota is provided in this section. It will be emphasized in the following 3.2.2.5.
10.Comment:
General recommendation for Section 1.2:
The section offers a broad and informative overview of current lifestyle and pharmacological strategies for metabolic disease management. However, given the manuscript’s core focus on microbiota modulation, I strongly recommend reinforcing the microbiota–host interaction perspective throughout both subsections. Linking therapeutic interventions to microbial modulation will enhance the manuscript's coherence and thematic integrity.
Response:Thank you for your proposal. We carefully examined the manuscript itself and added more relevant treatment methods related to the gut microbiota that were in accordance with the theme. We will elucidate more realistically in Section 3.2.2.4 of the following article.
11.Comment:
2.1.1. Therapeutic Approaches for Diseases of the Digestive System
- The section effectively highlights links between microbiota and digestive diseases such as Crohn’s disease, colitis, and gastrointestinal side effects of cancer therapy.
- It accurately references microbiota and metabolite effects in cancer and mucositis therapy.
- However: The first example focuses on autism spectrum disorders (ASD), which are neurological rather than digestive conditions. Although the point is valid, it may confuse the subsection’s focus. It would be more appropriate to relocate this discussion to the neurological section.
- Greater depth is needed regarding specific therapeutic modulation, such as:
-Which probiotic strains have demonstrated efficacy?
-Which fermented compounds exert beneficial effects?
-What is the role of the gut barrier–microbiota axis?
Response:Thank you for your suggestion. Regarding your first suggestion, we have revised the content of the manuscript. Firstly, to avoid straying from the topic, we removed the section related to "autism spectrum disorder" from the article and added more diseases related to the digestive system, as well as the relationship between the gut microbiota and digestive system diseases.
In addition, in response to your second suggestion, we have added probiotic strains that have been proven effective in treating digestive system diseases and the role of the intestinal barrier - microbial axis. The importance of the gut microbiota was further emphasized.
12.Comment:
2.1.2. Therapeutic Approaches for Diseases of the Neurological System
- This section remains overly general and does not address specific neurologicalconditions such as Alzheimer’s disease, depression, or Parkinson’s disease.
- It would be helpful to include clinical or preclinical studies showing tangible impacts of microbial modulation on neurological health, such as Lactobacillus rhamnosus in anxiety and stress models.
- The section lacks a concluding paragraph to summarise therapeutic implications and the translational challenges involved.
Response:We appreciate this comment very much.In response to your proposal, we have revised the content of the article by adding the regulatory efficacy and mechanism of Lactobacillus rhamnosus on models such as anxiety and stress. In line 278 on page 7 of Section 3.1.2, it provides relevant literature support for the treatment of mental system diseases from the perspective of gut microbiota. And on line 291 on page 7 of Section 3.1.2, the potential of treating neurological diseases through the gut microbiota at present has been added.
13.Comment:
2.1.3. Therapeutic Approaches for Diseases of the Urinary System
- Requires more specificity and organisation: it discusses urinary tract infections, kidney transplantation, chronic kidney disease, and urological cancers without a clear hierarchy or thematic separation.
- It is recommended to divide the section into two thematic parts:
(1) Urinary tract infections;
(2) Chronic kidney disease and urological cancers. - The section would benefit from concrete therapeutic proposals, such as the use of targeted probiotic strains to reduce infection recurrence or to mitigate renal damage via microbial toxin modulation.
Response:
Thank you for your suggestions. In order to accurately clarify the hierarchical structure of this section, we have divided the content of this section into two parts.
The first part elaborates on the level of urinary tract infection. It has increased the mechanism of action of probiotics in urinary tract infections and their potential health benefits. The additional content is on line 304 on page 7 of 3.1.3.1 of the article.
The second part elaborates on chronic kidney disease and urinary system cancers, and adds the potential role of probiotics in the treatment of chronic kidney disease and enhancing immunotherapy. The additional content is on line 323 on page 7 of Section 3.1.3.2 of the article. All the added content is displayed in red font.
14.Comment:
General recommendation for Section 1.2:
The section offers a broad and informative overview of current lifestyle and pharmacological strategies for metabolic disease management. However, given the manuscript’s core focus on microbiota modulation, I strongly recommend reinforcing the microbiota–host interaction perspective throughout both subsections. Linking therapeutic interventions to microbial modulation will enhance the manuscript's coherence and thematic integrity.
Response:
Thank you for your suggestion.
By resetting the subheadings, the content of this section has been strengthened, and application examples of probiotics and other microbiota in treating diseases have been added, providing a more solid foundation for the treatment of diseases by intestinal microbiota.
15.Comment:
2.2.1.1. Obesity and T2DM
- The narrative conflates obesity and T2DM without clearly distinguishing their respective microbial profiles and pathophysiological mechanisms.
- Recent references concerning Faecalibacterium prausnitzii, Roseburia spp., and their protective roles are lacking.
- The contribution of increased intestinal permeability to insulin resistance should be more explicitly highlighted.
- Recommendation: Separate the discussion of obesity and T2DM into distinct subsections or clearly demarcated paragraphs. Incorporate recent clinical evidence and strengthen the discussion of immunometabolic mechanisms involved.
Response:
Thank you for your detailed suggestions. We have optimized this section.
First of all, in order to avoid confusing obesity with T2DM, we have divided this section into two parts: 3.2.1.1 Obesity and 3.2.1.2 T2DM. Secondly, we discussed the main mechanism of obesity in line 361 on page 8 of Section 3.2.1.1 of the article.
In addition, we discussed the influence of intestinal permeability on insulin resistance in line 379 on page 9 of Section 3.2.2.2 of the article. These mechanisms reveal the crucial role of the gut microbiota in metabolic diseases. The modified content is marked in red in the article.
16.Comment:
2.2.1.2. Hyperlipidaemia
- The subtitle should specify that the section primarily focuses on NAFLD as a model of dyslipidaemia.
- Bacterial species such as Klebsiella pneumoniae and Cholera spp. are mentioned without explanation of their specific roles or links to metabolic profiles.
- The involvement of gut microbiota in hepatic gene regulation (e.g., FXR, SREBP-1c) is not discussed.
- Recommendation: Clarify that hypertriglyceridaemia is addressed via the gut–liver axis, and strengthen the connection between microbial metabolites (e.g., ethanol, LPS) and hepatic lipogenesis.
Response:
Thank you for your professional guidance.
Regarding your first suggestion. The content of this section is divided into two sections for separate elaboration, namely 3.2.1.3 Hyperlipidemia and 3.2.1.4 NAFLD.
Regarding your second suggestion, the influence of Klebsiella pneumoniae and cholera bacteria on the development of NAFLD is expounded in line 413 on page 9 and line 424 on page 19 of Section 3.2.1.4.
Regarding your third suggestion, the genes related to the regulation of liver metabolism by the gut microbiota were profoundly discussed in line 403 on page 9 of Section 3.2.1.3 of the article. It reveals the role of intestinal bacteria at the molecular level.
Regarding your fourth suggestion, it is elaborated in 3.2.1.3, strengthening the connection between microbial metabolites and liver adipogenesis.
17.Comment:
2.2.1.3. Hyperuricaemia and Gout
- The section lacks deeper mechanistic insights into how microbiota alterations exacerbate hyperuricaemia (e.g., via intestinal permeability, systemic inflammation).
- It does not specify whether findings are consistent across human, animal or in vitro studies.
- There is no mention of whether probiotic-based therapies exist for the management of hyperuricaemia.
- Recommendation: Add a paragraph providing functional integration (e.g., microbiota–inflammation–renal protection axis), and relate it to current evidence on probiotic or prebiotic interventions.
Response:
Thank you for your inspiration.
In line 449 on page 10 of Section 3.2.1.5 of this article, the LPS produced by Gram-negative bacteria is described, which leads to a series of chain reactions in the body. It has been proved that microorganisms can aggravate hyperuricemia through their metabolites.
Regarding the relevant probiotic therapy, we will elaborate in detail on line 490 on page 11 of Section 3.2.2.2.
18.Comment:
2.2.2.1. Dietary Treatment
- This section is overly brief and general; no specific examples of dietary patterns (e.g., polyphenol-rich diets, Mediterranean or ketogenic diets) are provided despite their known impact on the microbiota.
- Detrimental dietary patterns (e.g., high-fat or high-sugar diets) are not critically addressed as a point of contrast.
- The section would benefit from more recent references (2023–2024), particularly clinical studies or meta-analyses on dietary interventions targeting the gut microbiota.
- Suggestion: Include specific examples of fermentable fibres and relate them to studies in human or animal models of metabolic disease.
Response:
Thank you for your proposal.
Regarding your first proposal, we elaborated on the impact of the Mediterranean diet and the ketogenic diet on the microbiome on page 11, line 474 of Section 3.2.2.1 of the article. These examples illustrate that different types of diets have different effects on the function of the gut microbiota.
Regarding your second suggestion. We elaborate on the dietary patterns of obese patients on line 485 on page 11. Additionally, the third suggestion of fermentable fibers has been added, revealing the need for deeper exploration and research in the field of fermentable fibers in the future.
19.Comment:
2.2.2.2. Probiotics, Prebiotics and Postbiotics Treatment
- Although hyperuricaemia is mentioned, the clinical evidence for other metabolic disorders is not integrated in a systematic manner.
- The section blends physiological information with studies of varying evidence levels (in vitro, murine, human), without clear distinction.
- Potential limitations of postbiotics, including safety or bioavailability concerns, are not discussed.
- Suggestion: Consider adding a comparative table summarising the effects of selected probiotic strains versus known postbiotics. Include a critical reflection on the synergy between these compounds and dietary components.
Response:
Thank you for the proposal.
Regarding your first suggestion. We elaborated on line 499 on page 11 of Section 3.2.2.2 of the article, connecting with other clinical evidence related to metabolic diseases and using multiple clinical studies to prove the availability and effectiveness of probiotics.
Regarding the second suggestion, we separate the clinical research and the mouse research within the paragraph to avoid confusion.
Regarding your third proposal, we separate probiotics from prebiotics and postbiotics as 3.2.2.2 and 3.2.2.3. In line 558 on page 12 of Section 3.2.2.3, the potential limitations of postbiotics were carefully explored, indicating that the research on postbiotics needs to be more in-depth to address their potential risks and optimize their application value.
In response to your fourth proposal, we have added Table 1 (page 14), whose content includes probiotics, postbiotics, and comparisons between human experiments and mouse experiments.
20.Comment:
2.2.2.3. Drug Treatment
- While microbiota-modulating effects of drugs are mentioned, a direct cause–effect relationship between microbiota changes and clinical outcomes is not consistently supported.
- The distinction between direct pharmacological effects and microbiota-mediated mechanisms is unclear.
- Recent literature examining longitudinal effects of drug treatments on the human microbiome (e.g., cohort studies or RCTs) should be incorporated.
- Suggestion: Expand this section to include data on interindividual variability in treatment response based on baseline microbiota composition, a key element in personalised medicine.
Response:
Thank you for your professional guidance.
By reading the literature, we described the direct effects of the drugs, such as liraglutide, etc., which have no obvious effect on the gut microbiota, in line 615 on page 13 of Section 3.2.2.5 of the article. Meanwhile, examples of the baseline characteristics of the intestinal microbiota affecting the therapeutic effect were added. On line 602 on page 13 of the article, it was proved that it is a key factor in individualized medicine.
21.Comment:
3.1 – TCM for metabolic diseases
- The introductory paragraph of this section is overly general and lacks up-to-date key references regarding microbiota–TCM interactions in humans.
- A clearer delineation of the principal mechanism of action to be discussed in each subsection would be appreciated (e.g., effects on SCFAs, intestinal barrier, inflammatory pathways).
- It is recommended to incorporate clinical examples or early human trials, as the current focus is largely preclinical.
Response:
Thank you for the proposal.
In response to your proposal, I have incorporated the content about the interaction between the human microbiota and TCM into different paragraphs, providing detailed descriptions of each disease within the paragraphs. The mechanism of action of its TCM is described in detail in each subsection.
22.Comment:
3.1.1 – TCM for obesity and T2DM
- Mechanistic depth is lacking: the section does not detail how these compounds modulate the gut microbiota or which bacterial taxa are involved.
- It is advised to add references to recent studies documenting the direct interaction between TCM compounds and gut microbiota in obesity/T2DM models (e.g., the role of Akkermansia muciniphila with berberine).
- Statements such as “TCM can treat obesity in many ways...” are overly general and lack mechanistic support or specific citations.
Response:
Thank you for your detailed proposal.
In this section, in order to avoid confusing Obesity with T2DM, this is divided into two parts, namely 4.1.1 T2DM and 4.1.2 Obesity.
Regarding your first suggestion, in line 641 on page 18 of Section 4.1.1, a detailed elaboration is provided on the regulation of gut microbiota by different TCM compounds for the treatment of T2DM.
Regarding the second suggestion, the role of TCM compounds and gut microbiota in obese patients is elaborated in detail in line 675 on page 18 of Section 4.1.2.
Regarding the third suggestion, some sentences in the article have been modified.
23.Comment:
3.1.2 – TCM for NAFLD
- More detail is needed regarding the effects of these compounds on gut microbiota. For instance, how do these plants influence bacterial ethanol production, SCFA profiles, or hepatic gene expression?
- The clinical relevance of the compounds is unclear: is there evidence of human use or are findings limited to murine models?
- This subsection could benefit from an additional table or infographic summarising mechanisms and active ingredients of the TCM agents discussed.
Response:
Thank you for your guidance.
Firstly, since NAFLD is one of the dyslipidemia models, NAFLD is regarded as the main focus of this section, and the title of the article is changed to "Hyperlipidemia".
In response to your suggestions, in this section, we have added that TCM molecules influence the gut microbiota through different mechanisms (such as the genetic level and the molecular level) to treat hyperlipidemia and NAFLD, etc. The additional content is on line 703 on page 19 of Section 4.1.3.
In addition, this section can benefit from Table 2 (page 20, line 763), which summarizes the TCM and active ingredients discussed.
24.Comment:
3.1.3 – TCM for hyperuricaemia and gout
- This part could be strengthened by including microbial differences in gout patients, such as shifts in butyrate-producing bacteria or pro-inflammatory flora.
- Indicate whether there are clinical trials or current limitations to the real-world therapeutic application of TCM in this context.
- Include how fermented TCM products could enhance the bioavailability of compounds such as diosgenin.
Response:
Thank you for your guidance.
In this section, we have strengthened the research on the limitations of the therapeutic effects of TCM on hyperuricemia and gout.The added content is on line 751 on page 20
25.Comment:
3.2 – Extraction of active ingredients of TCM
- Lack of molecular or functional focus: While this section outlines extraction techniques, it does not explicitly connect them to the therapeutic functionality of the extracted compounds. It would be valuable to include examples linking bioactive compound types (e.g., flavonoids, alkaloids, polysaccharides) with relevant metabolic or immunomodulatory actions on the microbiota.
- Absence of selection criteria: Although multiple methods are described, there is no clear rationale for choosing one over another depending on compound type or intended application, which may reduce the practical utility for researchers.
- Poor integration with the manuscript’s main theme: The section is insufficiently linked to the central focus of the manuscript, which is the synergy between fermented herbal bioactives and probiotics. It would be beneficial to explain how the chosen extraction method may influence the bioavailability and fermentability of compounds by probiotics.
- Superficial technical language: Certain terms, such as “semimimetic extraction”, are not well defined or contextualised, which could confuse non-specialist readers.
Response:
Thank you for your sincere suggestion.
In response to your first suggestion, we have revised the content of the manuscript. In line 793 on page 21 of Section 4.2 of the article, relevant contents such as how the molecules of TCM compounds are transformed after fermentation and how their therapeutic effects are improved (such as flavonoids, etc.) have been added.
In response to your second suggestion, we have added relevant knowledge in the article (21 pages, 794 lines) that different extraction methods have different effects, expanding the professional knowledge related to the manuscript.
In response to your third suggestion, we have modified and deleted some superficial language.
26.Comment:
3.3 – TCM Fermentation Methods
- Lack of a clear conceptual framework: Although the section refers to the benefits of fermentation, it does not sufficiently develop the biotechnological and microbiological context of these transformations, nor does it explicitly connect them to specific therapeutic goals.
- Limited integration with human metabolism or gut microbiota: The discussion lacks direct links between compound transformations and their effects on gut microbiota or host metabolic pathways—essential considerations in a manuscript focused on metabolic therapy.
- Imprecise language in some statements: Phrases such as “may improve efficacy” or “can generate new effects” are too vague. It is recommended to use evidence-based language such as “has been shown to...” or “increases the concentration of...”.
3.3.1 Traditional Fermentation
- Specific examples of traditionally fermented products and their clinical applications in TCMshould be included.
- Recent references comparing the microbial activity of solid-state versus liquid-state fermentation are lacking.
Response:
Thank you very much for your careful guidance to us.
In response to your suggestions, we have made modifications to Part 4.3 and 4.3.1. The inaccurate statements in the text have been deleted and modified.
In the modifications of the following sections, pay attention to the connection between the compounds produced and the host metabolism.
In 4.3.1, for your first suggestion, specific examples of traditional fermentation products and their applications have been added on line 814 on page 22.
Regarding your second suggestion, on line 825 on page 22, a comparison of the microbial activities in solid-state and liquid fermentation was made.
27.Comment:
3.3.2 Probiotic Fermentation
- This section would benefit from improved contextualisation in terms of the probiotic species employed, fermentation conditions, and the nature of fermentation-derived products.
- Batch-to-batch consistency and microbial standardisation are not addressed, yet these are critical for therapeutic reproducibility.
Response:
Thank you for your guidance.
In line 847 on page 22 of Section 4.3.2 of the text, we have added the elaboration on the production, screening and standardization of probiotics. Such as requiring the screening of the characteristics of the strain itself and ensuring the final safety assessment, etc.
28.Comment:
3.3.2.1 Enhancing Active Ingredients
The discussion should contrast the actual bioavailability of bioactive compounds post-fermentation, not merely their increased concentration.
Response:
Thank you for your guidance.
Regarding your suggestion. In line 874 on page 23 and line 883 on page 23 of Section 4.3.2.1 of the article, examples of the increased practical utilization of active ingredients after probiotic fermentation have been added.
29.Comment:
3.3.2.2 Reducing the Toxicity
It would be valuable to include molecular detoxification mechanisms (e.g., specific enzymatic actions), rather than only reporting the final effect.
Response:
Thank you for your suggestion.
In response to your suggestion, the relevant molecular mechanisms and examples of detoxification of certain components in TCM after probiotic fermentation have been added on line 891 on page 23 of Section 4.3.2.2 of the article.
30.Comment:
3.3.2.3 Generating New Compounds
Further details are needed regarding the chemical nature and pharmacodynamic properties of newly generated compounds, and how these may be applied to metabolic diseases.
Response:
Thank you for your suggestion.
In response to your suggestion, the relevant molecular mechanisms and examples of detoxification of certain components in TCM after probiotic fermentation have been added on line 891 on page 23 of Section 4.3.2.2 of the article.
31.Comment:
3.3.3 Fermentation Strains
- The section is largely descriptive; a more comparative approach is encouraged, including strain selection criteria (e.g., pH tolerance, enzymatic profiles, synergy with phytochemicals).
- The strain-specific immunomodulatory roles of certain probiotics are not discussed but would be relevant to the proposed therapeutic scope.
- Integration with other sections should be improved (e.g., highlighting how specific strains relate to the molecular mechanisms discussed later in the manuscript).
Response:
Thank you for your careful guidance.
In response to your suggestion, we have added the selection criteria for probiotics when choosing fermentation products in line 926 on page 24 of Section 4.3.3 of the article. Including PH tolerance, etc.
In line 925 on page 25, it has been added that by comparing the different characteristics of the strains and their interactions with the components of TCM, the therapeutic effect can be better exerted.
32.Comment:
3.4.1 TCM as a Natural Prebiotic
- Key references regarding dose-dependent effects and clinical outcomes are missing.
- The heterogeneity of the human microbiome, potential adverse effects, and therapeutic limits are not addressed.
- A clearer distinction between in vivo and in vitro effects is needed, specifying whether data derive from animal models, human trials, or bacterial cultures.
Response:
Thank you for your suggestion.
Regarding your first suggestion, in line 977 on page 25 of Section 4.4.1 of the article, the literature support related to the regulatory effect of TCM on the gut microbiota and the dose-effect relationship of clinical efficacy within a certain dose range has been added, illustrating the application of dose-dependent medicine in TCM.
In response to your second suggestion, we have added knowledge content about the relevant limiting factors such as the heterogeneity of the human microbiome and the expected individual differences in related clinical outcomes in line 985 on page 25 of the article.
Regarding your third suggestion. We explored the limitations of current experiments focusing on human experiments. When adopting traditional Chinese medical therapies, multiple factors need to be considered. (25 pages, 989 lines)
33.Comment:
3.4.2 TCM and the Intestinal Mucosal Barrier
- The discussion of molecular mechanisms remains superficial; references to specific tight junction proteins such as ZO-1 or occludin (although mentioned later) should be incorporated here.
- Adding the relationship between barrier integrity and inflammatory markers would enrich this section.
- Clinical trial data demonstrating a correlation between mucosal repair and metabolic improvements are notably absent.
Response:
Thank you for your careful guidance.
In response to your first suggestion, we have added relevant knowledge points about tight junction proteins on page 26, line 1019 of the article.
In response to your second suggestion, we have added literature support and content related to inflammatory markers and the integrity of intestinal barrier function on page 26 and 999 lines.
Regarding your third suggestion, we have elaborated in the paragraph on the improvement of metabolism after mucosal repair and the related content of improving metabolic diseases.
34.Comment:
4.Challenges and Future Directions
- Lack of critical prioritisation: While many challenges are listed, they are not ranked or weighed in terms of impact or urgency. The reader is left uncertain about the most significant bottlenecks to clinical translation.
- Scarce quantitative evidence: Despite broad coverage, no concrete examples or key studies are cited to support some major claims (e.g., recurrence of dysbiosis post-FMT or scalability limitations of solid fermentation extracts).
- Limited molecular depth: Although “multi-omics” technologies are mentioned, no specific biomarkers or metabolic pathways are discussed that could guide microbiota-based precision medicine.
- Lack of concrete innovation proposals: While there is general mention of “interdisciplinary collaboration and technological innovation”, it would be beneficial to propose specific models or platforms that foster such collaboration (e.g., international consortia, metagenomic data-sharing networks, or clinical biobanks with microbiota–host phenotyping).
Response:
Thank you for your suggestion.
In Section 5, page 28, page 1130 of the article, we have added relevant knowledge such as the research on multi-omics technology - metagenomic technology, and conducted relevant discussions on micro RNA.
Thank you again for the suggestion.

Round 2
Reviewer 1 Report
Comments and Suggestions for Authors
The revised version of the manuscript has addressed all the recommendations and its quality has increased considerably.